# Analysis of cranial tenocyte heterogeneity reveals a role for Wnt signaling in tendon attachments

Arul Subramanian[1,*], Pavan K. Nayak[1,*], Cameron L. Miller[2], Daniel B. Dranow[1], Ryan R. Roberts[3], J. Gage Crump[3] and Thomas F. Schilling[1,‡]

## ABSTRACT

Tenocytes secrete the extracellular matrix (ECM) of tendons and ligaments in response to mechanical forces from the muscles and bones to which they attach. Although these tissues are often injured and weaken with age, we know little about the genetic mechanisms controlling their development or maintenance. Through single-cell RNA sequencing (scRNA-seq) of connective tissues in the embryonic zebrafish head, we identify distinct subpopulations of developing tenocytes and ligamentocytes. Spatially distinct transcriptional cell signatures, particularly for ECM genes, correlate with the type of tendon/ligament (i.e. longer load-bearing skeletal attachments versus soft tissue attachments) as well as tenocyte locations within tendons (i.e. skeletal entheses versus myotendinous junctions). Combinatorial *in situ* analyses confirm spatial co-expression of genes defining many of these subsets of tendon or ligament cells. From pathway analysis, the scRNA-seq data also suggest a role for canonical Wnt signaling in tenocyte development. Genetic and pharmacological Wnt manipulations alter tenocyte aggregation and cause ectopic cranial muscle attachments. These findings reveal previously unappreciated spatial and functional heterogeneity in tenocytes during embryogenesis and define a role for Wnt signaling in attachment patterning and morphogenesis.

KEY WORDS: Tenocytes, Ligamentocytes, Enthesis, Myotendinous junction, Transcriptional heterogeneity, Zebrafish

## INTRODUCTION

Tendons and ligaments are extracellular matrix (ECM)-rich connective tissues that interconnect muscles and bones and transmit contractile forces from muscle activity. Tendon fibroblasts (tenocytes) secrete collagens and other structural ECM proteins that organize into tendon fibrils (Eisner et al., 2022; Subramanian and Schilling, 2015). Few studies have examined heterogeneity in tenocyte gene expression during development, and none has described how this influences the variety in structure and function of tendons (i.e. tendons attaching to hard versus soft tissue) (Maeda et al., 2010; Schweitzer et al., 2001; Shukunami et al., 2006; Zhang and Wang, 2013). For example, the ECM composition and organization of a tendon depends on its interface with muscle-myotendinous junction (MTJ) or bone/cartilage (enthesis) (Karlsen et al., 2022; Schwartz et al., 2012; Zelzer et al., 2014). RNA sequencing (RNA-seq) of MTJ and entheseal cells from adult tendons have shown that their tenocytes differ from those populating the tendon body (mid-substance) (Kult et al., 2021; Petrany et al., 2020; Zhang et al., 2023). These differences correlate with the mechanical forces associated with their stiffness and elasticity (Lipp et al., 2023; Schwartz et al., 2013). Thus, it is important to understand how tenocyte transcriptional state relates to tendon diversity, stiffness of attachment interface and muscle function to improve treatment of tendon injuries.

Vertebrate tenocytes are specified during development by transcription factors (TFs) including Scleraxis (Scx), Early growth response 1/2 (EGR1/2) and Mohawk (Mkx) (Anderson et al., 2006; Lejard et al., 2011; Schweitzer et al., 2001). The basic helix-loop-helix TF Scx is expressed in embryonic and adult tenocytes and directly regulates transcription of tendon ECM genes (Paterson et al., 2020; Shukunami et al., 2018; Subramanian and Schilling, 2015). Although *Scx* is widely expressed by all tenocytes, loss of *Scx* in mice selectively affects load bearing tendons and double mutants in *scxa* and *scxb* orthologs in zebrafish disrupt medial cranial tendons, suggesting that Scx is not necessary for development of all tendons (Kague et al., 2019; Murchison et al., 2007). TFs of the Cyclic-AMP-responsive element-binding (CREB) and Early B-cell factor (EBF) families also activate Scx expression in tenocyte progenitors, and Transforming growth factor, Beta (TGFβ) signaling is required for their patterning and maintenance (Niu et al., 2024; Pryce et al., 2009). Most studies have focused on elongated limb tendons attached to bones, such as the Achilles and supraspinatus, and have profiled heterogeneity in tenocyte transcriptomes and proteomes from individual tendons. However, few have compared tenocytes within and between multiple tendons or examined the development of tendons that attach muscles to soft tissues such as eyes and other muscles (Benjamin et al., 2008; Fu et al., 2023; Karlsen et al., 2022; Kult et al., 2021; Steffen et al., 2023; Yan et al., 2022). Although cranial tenocytes and ligamentocytes largely originate from cranial neural crest (NC) cells (unlike trunk and limb tenocytes which originate from mesoderm), they express similar suites of factors, including Scx, Mkx, Collagen 1 (Col1), Thrombospondin 4 (Thbs4) and Tenomodulin (Tnmd) (Brent and Tabin, 2004; Grenier et al., 2009; Havis et al., 2014; Noden and Trainor, 2005; Nödl et al., 2022; Schmidt et al., 2013; Subramanian and Schilling, 2015; Yamamoto and Abe, 2020). We chose to study cranial tendon heterogeneity in zebrafish as their cranial tendons become functional during early stages of embryogenesis, which allows for *in situ* staining for gene expression and live imaging of their development.

Most previous studies have profiled gene expression in mature, fully differentiated tenocytes, and we have little knowledge of how

[1]Department of Developmental and Cell Biology, University of California, Irvine, CA 92617, USA. [2]Center for Complex Biological Systems, University of California, Irvine, CA 92617, USA. [3]Department of Stem Cell Biology and Regenerative Medicine, Keck School of Medicine, University of Southern California, Los Angeles, CA 92617, USA.
*These authors contributed equally to this work

‡Author for correspondence (tschilli@uci.edu)

A.S., 0000-0001-8455-6804; P.K.N., 0000-0002-4360-6729; C.L.M., 0009-0005-7407-0698; D.B.D., 0000-0002-5685-3833; J.G.C., 0000-0002-3209-0026; T.F.S., 0000-0003-1798-8695

and when tendon heterogeneity arises during embryogenesis (Havis et al., 2014). In zebrafish, embryonic onset of muscle contraction regulates tendon ECM organization, tenocyte morphogenesis and expression of force-responsive genes, in part through TGFβ-dependent mechanotransduction (Malbouyres et al., 2022; Nayak et al., 2025; Subramanian et al., 2018, 2023). Since the forces transmitted by tendons vary depending on the type of attachment (cartilage, bone or soft tissue) an important open question is how the differences in force influence tenocyte heterogeneity.

Here, we profile cranial tenocyte heterogeneity in embryonic zebrafish using single-cell RNA sequencing (scRNA-seq) on *scxa:mCherry*-sorted cells. We find distinct classes of embryonic tenocytes associated with individual tendons or ligaments that reflect spatial patterning as well as function. These subtypes vary by ECM expression profiles [e.g. fibrillar, basement membrane or fibril associated collagens with interrupted triple helices (FACIT) collagens] that correlate with their functional demands. In addition, regional heterogeneity within tendons emerges early during development. We confirm spatial specificity of many of these tenocyte markers *in vivo* using *in situ* hybridization chain reaction (*is*HCR) (Choi et al., 2010). Pathway analysis reveals a potential role for canonical Wnt signaling in embryonic cranial tendon development. Consistent with this hypothesis, genetic and pharmacological manipulations demonstrate roles for Wnt signaling in MTJ formation and the migration of a subpopulation of cranial tenocytes associated with medial tendons.

## RESULTS

### scRNA-seq of embryonic cranial tenocytes identifies distinct populations

We obtained cranial tenocytes for analysis from the severed heads of 72 h post-fertilization (hpf) *Tg(scxa:mCherry)* embryos, which express mCherry driven by *scxa* regulatory elements (Subramanian et al., 2018). By 72 hpf, zebrafish had developed functional jaw muscles with tendons at their attachment sites (Fig. 1A,B; Table 1). Cells were dissociated using a cold-protease protocol to suspend transcription and minimize cell stress artifacts before cell sorting (Subramanian et al., 2025). mCherry+ cells from three independent replicates were subjected to fluorescence activated cell sorting (FACS) and processed for 10x Genomics scRNA-seq. Among the sorted cells, 67% expressed *mCherry*, 44% expressed *scxa* and 37% co-expressed mCherry and *scxa* (Fig. S1). We used Seurat to integrate all three replicates and, after unsupervised clustering, obtained 16 clusters (Fig. 1C-F; Table S1). Cells from all three replicates contributed to each cluster, suggesting we captured even distributions of cell populations across replicates (Fig. 1D). Nine *scxa:mCherry*-positive clusters showed a higher score for a 'Tendon core module' of classical tenocyte markers [*scxa*, *mkxa*, *tnmd*, *thbs4b* and *tenascin b* (*tncb*)] (Fig. 1E-G; Table 2) (see Materials and Methods). Other cell types present included chondrocytes, macrophages and keratinocytes, presumably contaminants during sorting (Fig. 1F). Consistent with their cranial NC origins, tenocyte clusters were resolved in the UMAP by genes associated with anterior-posterior (A-P) and dorsal-ventral (D-V) patterning of pharyngeal arch NC-derived mesenchyme [e.g. homeobox (hox) genes, distal-less homeobox (dlx) genes, *hand2*, *barx1*] (Fig. 1F; Fig. S2).

Using *is*HCR, we observed spatially distinct expression patterns for tenocyte markers *thbs4b*, *scxa* and *mkxa* (Fig. 1H). *thbs4b* expression was stronger in the MTJs of the HH-HH tendon, mc-AM tendon, and extraocular muscle tendons (EOMTs) (sm-IO, sm-IR), but lower in the ch-SH tendon (elongated MTJ) and mc-pq and pq-hs ligaments (Fig. 1A,B,H). While most tenocytes with

strong *thbs4b* expression also strongly expressed *scxa*, the EOMT tenocytes showed lower *scxa* expression compared to *thbs4b* expression or mCherry signal, and *mkxa* expression primarily localized to the ch-HH tendon, mc-pq, pq-hs ligaments, and HH-HH tendon (Fig. 1A,H). These data suggest that transcriptional differences distinguish spatially distinct tendons during zebrafish embryogenesis.

To resolve tenocyte heterogeneity between individual developing cranial tendons, we performed unsupervised clustering with increased cluster resolution to account for population expression specificity, selected marker genes from individual clusters and probed their expression in 72 hpf embryos using *is*HCR (Fig. 2A; Fig. S3A; Table S2). By performing gene expression colocalization together *in vivo* (double *is*HCR with *scxa* or other tendon markers) and *in silico* (visualized using Nebulosa, which we reasoned would reflect the co-expression observed in *is*HCR) we identified ten higher resolution clusters belonging to individual cranial tendons/ligaments (Fig. 2B; Figs S3A and S4; Table S2) (Alquicira-Hernandez and Powell, 2021). For example, *is*HCR showed that *scxa*+ tendons of the mandibular arch (mc-IMA and mc-IMP) also expressed *nuclear receptor subfamily 5 group A member 2* (*nr5a2*) (Fig. 2G). *scxa*+/*nr5a2*+ co-expressing cells from cluster 6 were therefore assigned as mc-IMA/IMP tenocytes (Fig. 2G; Fig. S4). UMAP gene expression was then corroborated with other *is*HCR probe patterns to further narrow down cluster specificity, i.e. the *scxa*+/*nr5a2*+ cells of clusters 20 and 36 were ruled out as mc-IMA/IMP tenocytes, since these two clusters also expressed *fibronectin type III domain containing 1* (*fndc1*). *is*HCR showed that *fndc1* was expressed in several distinct subsets of cartilage- and joint-associated tenocytes, including the mc-mc and mc-pq joints, IMP-IH and ch-SH tendons, and the mc-pq, pq-hs ligaments, but not the mc-IMA/IMP tenocytes (Fig. 2H; Fig. S4) (Chen et al., 2023).

Other cluster-tendon/ligament annotations included: *mkxb*+/*scxa*+ and *tnmd*+/*scxa*+ tenocytes of the ch-SH belonging to cluster 35 (Fig. 2C; Figs S3B-F and S4); *scxa*+/*elastin microfibril interfacer 3a* (*emilin3a*)+ cranial joint cells of the mc-mc and ch-hs belonging to cluster 32 (Fig. 2E) (Truong et al., 2024); *scxa*+/*iroquois homeobox 1b* (*irx1b*)+ cells of the EOMTs belonging to cluster 19 (Fig. 2F); *thbs5*+ [also known as *cartilage oligomeric matrix protein* (*comp*)]/*tbx1*+ (a muscle marker) HH-HH tenocytes belonging to cluster 29 (*thbs5* was expressed in most cranial tenocytes) (Fig. 2I; Figs S3L-P and S4); and *scxa*+/*alx1*+ mc-mc (iml) ligamentocytes belonging to cluster 12 (*alx1* is essential for cranial NC cells of the frontonasal primordia) (Figs 1A and 2J; Fig. S4) (Pini et al., 2020).

In addition to these tendons/ligaments associated with the endochondral pharyngeal and neurocranial skeleton, which are well established at 72 hpf, we also identified tenocyte populations associated with the developing dermal skeleton (intramembranous ossification) (Fig. 2D). For example, *interferon induced transmembrane protein 5* (*ifitm5*), a transmembrane protein expressed by osteoblasts that aids in mineralization (Hanagata et al., 2011), was co-expressed with *scxa* in tenocytes at the tendon attachments of the operculum and cleithrum (cl-SH, cl-HYP, op-DO) (Fig. 1B), which plotted to cluster 30 in our UMAP (Fig. 2D; Fig. S4). *angiopoietin like 7* (*angptl7*) was expressed in most cranial tenocytes as well as marking tenocytes of the pectoral fins (Figs S3G-K and S4) based on co-expression of pectoral fin marker *T-box transcription factor 5a* (*tbx5a*) and represented by cluster 16 (Fig. 2K). These distinct expression patterns clearly show establishment of tenocyte transcriptional diversity associated with spatially distinct tendons and ligaments.

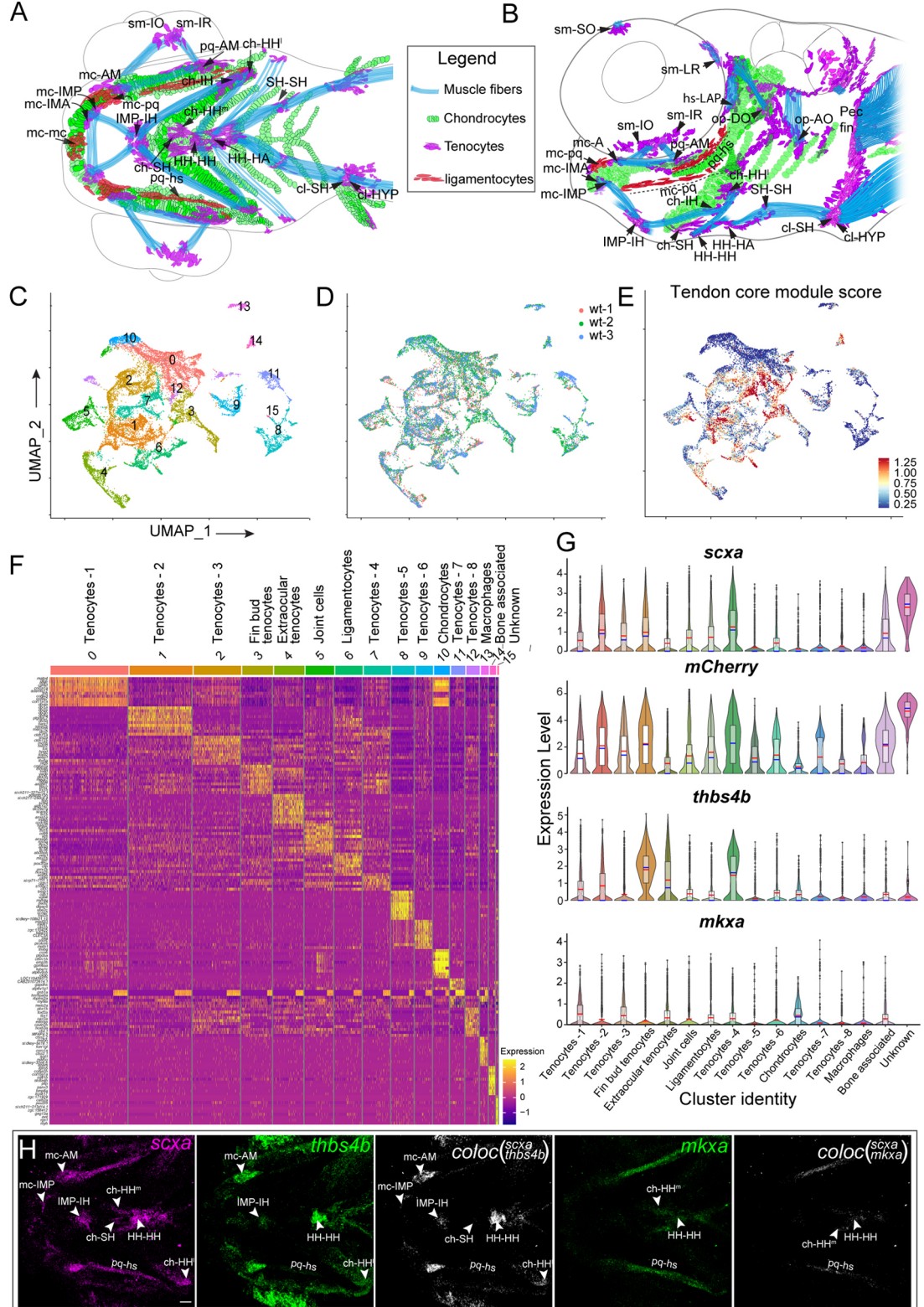

**Fig. 1. scRNA-seq reveals embryonic cranial tenocyte heterogeneity.** (A,B) Diagrams of ventral (A) and lateral (B) views of cranial musculoskeletal tissues in a 72 hpf zebrafish embryo showing tenocytes, ligamentocytes, muscle fibers and chondrocytes drawn from images of *Tg(scxa:mCherry; sox10:eGFP)* transgenics. (C-E) Unbiased clustering of single cell RNA-sequencing (scRNA-seq) from FAC-sorted tenocytes shown in a dimensionally reduced UMAP plot (C), an overlay of three biological repeats (D) and a feature plot of relative module expression scores of tendon core genes (E). (F) Heat map displaying expression of top 10 variable genes in each cluster. (G) Violin plots with box plot overlaps, depicting expression of key tendon genes (*scxa*, *thbs4b*, *mkxa*) and transgenic marker *mCherry* in tenocyte clusters (median and mean are depicted as horizontal blue and red lines, respectively). The box region comprises 50% of cells in the cluster, representing the interquartile range (IQR), while the bottom and top regions represent 25th and 75th percentile. The whiskers represent the spread of data outside the IQR within the normal variability of gene expression. (H) *is*HCR images showing spatial expression of *scxa*, *thbs4b*, *mkxa* and in ventral views of 72 hpf embryos including colocalization (arrowheads) of *scxa; thbs4b* and *scxa;mkxa* in tenocytes. Scale bar: 20 μm.

**Table 1. Cranial musculoskeletal anatomical names, abbreviations and associated clusters**

| Abbreviation | Name | Tendon/ligament | Markers | Cluster ID |
|---|---|---|---|---|
| sm-IO | Scleral mesenchyme-inferior oblique | Tendon | *irx1b/scxa* | 19 |
| sm-IR | Scleral mesenchyme-inferior rectus | Tendon | *irx1b/scxa, klf2b/scxa* (enthesis), *col22a1/scxa* (MTJ) | 19 |
| sm-SO | Scleral mesenchyme-superior oblique | Tendon | *irx1b/scxa* | 19 |
| sm-LR | Scleral mesenchyme-lateral rectus | Tendon | *irx1b/scxa* | 19 |
| hs-LAP | Hyosymplectic-levator arcus palatine | Tendon | | |
| op-AO | Opercle-adductor operculi | Tendon | | |
| op-DO | Opercle-dilator operculi | Tendon | | |
| pq-AM | Palatoquadrate-adductor mandibulae | Tendon | | |
| ch-HH$^m$ | Ceratohyal-hyohyoideus (medial) | Tendon | | |
| ch-HH$^l$ | Ceratohyal-hyohyoideus (lateral) | Tendon | | |
| ch-IH | Ceratohyal-interhyoideus | Tendon | | |
| ch-SH | Ceratohyal-sternohyoideus | Tendon | *mkxb, mkxb/scxa* (coloc), *thbs3a/scxa* (enthesis), *mkxb/thbs4b* (MTJ) | 35 |
| SH-SH | Sternohyoideus-sternohyoideus | Tendon | | |
| HH-HH | Hyohyoideus-hyohyoideus | Tendon | *comp/tbx1* (coloc) | 29 |
| HH-HA | Hyohyoideus-hyoidei adductores | Tendon | | |
| cl-SH | Cleithrum-sternohyoideus | Tendon | *ifitm5/scxa, ifitm5/scxa/klf2a* (enthesis) | 30 |
| cl-HYP | Cleithrum-hypaxial | Tendon | *ifitm5/scxa, ifitm5/scxa/klf2a* (enthesis) | 30 |
| mc-IMA | Meckels-intermandibularis anterior | Tendon | *nr5a2/scxa* | |
| mc-IMP | Meckels-intermandibularis posterior | Tendon | *nr5a2/scxa* | |
| mc-AM | Meckels-adductor mandibularis | Tendon | | |
| IMP-IH | Intermandibularis posterior-interhyoideus | Tendon | | |
| Pec fin | Pectoral fin | Tendon | *tbx5a/scxa* | 1, 16 |
| mc-pq; pq-hs | Meckels-palatoquadrate; palatoquadrate-hyosymplectic | Ligament | *fndc1/scxa/tnmd, nr5a2/scxa* | 36 |
| mc-mc | Meckels-meckels | Ligament/joint | *alx1/scxa* | 12 |
| mc-pq | Meckels-palatoquadrate | Joint | | |
| pq-hs | Palatoquadrate-hyosymplectic | Joint | | |

We also identified distinct *scxa*$^+$ cell populations based on genes not typically associated with tenocytes. These included clusters 33 and 37 that specifically expressed *macrophage expressed 1* (*mpeg1.1*). In a double transgenic line *Tg(mpeg1:NTR-eYFP; scxa: mCherry)*, we observed a few mpeg1:YFP-expressing macrophages dynamically interact with tenocytes from 48 hpf to 72 hpf during timelapse imaging (Fig. S5A,D,E; Movie 1). These two macrophage clusters were also positive for *mannose receptor, C type 1a* (*mrc1a*) and *mannose receptor, C type 1b* (*mrc1b*), the zebrafish orthologs of CD206. CD206$^+$ macrophages were recently shown to interact with tenocytes in mice (Bautista et al., 2023). Additionally, they were positive for cathepsin lysosomal

proteases *cathepsin S, ortholog 1* (*ctss1*), *cathepsin S, ortholog 2.1* (*ctss2.1*) and *cathepsin S, ortholog 2.2* (*ctss2.2*), as well as *platelet derived growth factor subunit Ba* (*pdgfba*), which are known to have functional roles in macrophage interactions with local ECM and tenocytes (Bautista et al., 2023; Brown et al., 2020) (Fig. S5E). *col4a5* was co-expressed with *scxa* in cluster 10, with *is*HCR suggesting that this population marked peri-tendinous and peri-ligamentous cells (Fig. S5A,B) (Taylor et al., 2011). Proliferating tenocytes were identified based on module score analyses for g2/m cell-cycle phase genes, which were highest in clusters 10 and 31 (see Materials and Methods; Fig. S5A,C; Table 2).

**Table 2. List of gene modules and associated genes used in module score calculation**

| Module | Genes |
|---|---|
| Tendon fibril collagen (TFC) | *col1a1a, col1a1b, col1a2, col5a1, col5a2a, col5a2b, col5a3a, col5a3b, col27a1a, col27a1b* |
| Tendon FACIT collagen (TFACIT) | *col9a1a, col9a1b, col9a2, col9a3, col12a1a, col12a1b, col14a1a, col14a1b* |
| Basement membrane collagen (BMC) | *col4a1, col4a2, col4a3, col4a4, col4a5* |
| Beaded filament collagen (BFC) | *col6a1, col6a2, col6a3, col6a4a, col6a4b* |
| Tendon proteoglycan (TP) | *lum, vcana, dcn, acana, acanb, bgna, fmoda, fmodb, prg4a, prg4b* |
| Tendon laminin (TL) | *lamb1a, lamc1, lama4, lamb1b, lamb2* |
| Tendon matrix metalloproteinase (TMMP) | *mmp2, mmp7, mmp9, mmp11a, mmp11b, mmp13a.1, mmp13b, mmp14a, mmp14b, mmp15a, mmp16a, mmp16b, mmp17a, mmp17b, mmp19* |
| Tendon tissue inhibitors of metalloproteinases (TTIMP) | *timp2a, timp2b, timp3, timp4.1, timp4.2, timp4.3* |
| Microfibril elastin (MFE) | *elna, elnb, fbn2, fbn3, emilin1a, emilin1b, emilin2a, emilin3a, emilin3b* |
| Lysyl oxidase (LO) | *loxa, loxl1, loxl2a, loxl2b, loxl3a, loxl3b, loxl4* |
| Enthesis (E) | *scxa, sox9a, klf2a, klf2b, itga10* |
| Myotendinous junction (MTJ) | *thbs4b, scxa, col22a1$^5$, comp, xirp2a, cilp, fermt2* |
| Proprioception (Prp) | *runx3, piezo1, piezo2b, ltbp1, ccn2a, ccn2b* |
| Tendon core module (TCM) | *scxa, mkxa, loxa, col1a1a, tnmd, col1a1b, thbs4a, thbs4b, tncb* |
| G2/M proliferating marker module | *smc4, tpx2, ckap2l, ect2, rangap1a, cdk1, tacc3, anln, ccnb2, nuf2, birc5a, kif11, kif20bb, tubb4b, anp32e, top2a, gtse1, cbx5, hmmr, cenpe, si:ch211-266i6.3, nusap1, nek2, cks1b, rangap1b, ctcf, ncapd2, hmgb2a, gas2l3, kif23, si:dkeyp-115e12.6, lbr, kif20ba, si:ch211-244o22.2, cdc20, zgc:65894, dlgap5, bub1, ttk, tmpob, tmpoa, aurka, si:ch211-69g19.2, selenoh, ckap5, ndc80, cks2, ube2c, aurkb, mki67, cdca8, zgc:153426* |

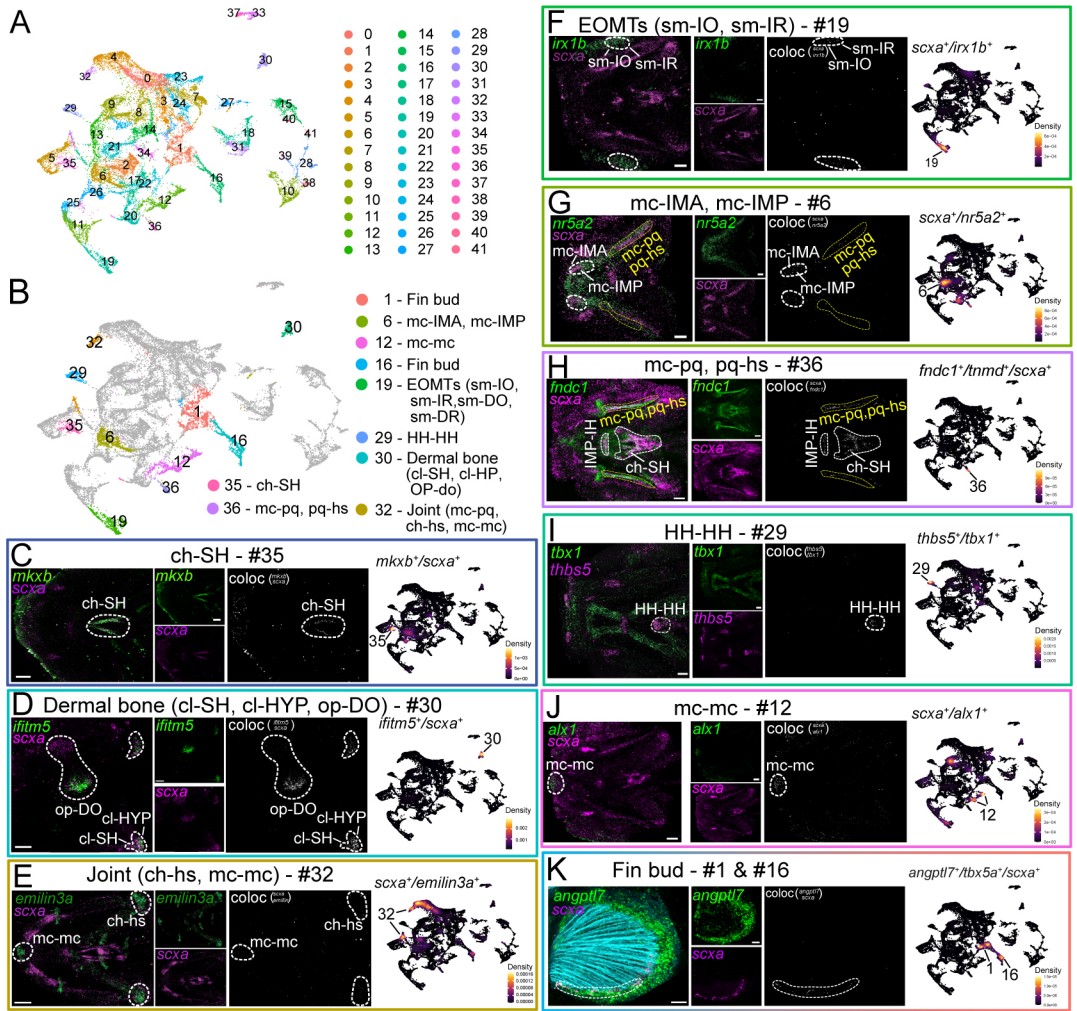

**Fig. 2. *is*HCR confirms spatial tenocyte heterogeneity.** (A,B) Subclustering of initial UMAP plots (A) showing ten subclusters validated using *in situ* hybridization chain reaction (*is*HCR) (B). (C-K) Ventral (C,E-J) and lateral (D,K) views of *is*HCR images of 72 hpf embryos showing cranial tenocytes expressing *scxa* and specific spatial markers (*thbs5, emilin3a, irx1b, nr5a2, fndc1, tbx1, mkxb, alx1, ifitm5* and *angptl7*). Monochrome panels show co-expression, and nebulosa plots show corresponding UMAP clusters. Dashed outlines show the tendon region marked by the gene expression associated with respective clusters depicted in the nebulosa heatmap. Scale bars: 30 μm (C,D,G-K); 20 μm (E,F).

## Spatially distinct tendons transcriptionally subcluster by their intra-tendon functionality

To determine if distinct transcriptional states of tenocytes match their intra-tendon functionality (e.g. MTJ, enthesis, mid-substance) at 72 hpf (Edahiro et al., 2023; Mathys et al., 2024) we subclustered individual spatially distinct tendon clusters at higher resolution (Materials and Methods). We selected clusters associated with softer (e.g. cluster 19 – EOMT) versus harder [e.g. cluster 35 – ch-SH (cartilage) or cluster 30 – cl-SH (dermal bone)] attachments (Figs 2A, 3A, 4A and 5A). Subclustering was followed by *is*HCR for respective cluster markers and known MTJ or enthesis markers to look for corresponding intra-tendon spatial expression domains.

Cluster 35 (ch-SH) subclustered into four populations: 'enthesis', with highest *thrombospondin 3a* (*thbs3a*) and enthesis marker *Kruppel like factor 2a* (*klf2a*) (Kult et al., 2021) expression; 'MTJ', with highest *thbs4b* and MTJ marker *col22a1* (Malbouyres et al., 2022) expression; and 'mid-substance enthesis'/'mid-substance MTJ', two clusters with *mkxb* expression and lower relative expression of *thbs3a, thbs4b* and *col22a1* (Figs 2C and 3A,F,K,O; Fig. S6A). *is*HCR supported the subclustering annotations, with *thbs3a* and *scxa* co-expressed at the ch-SH enthesis (Fig. 3B-E,G-J),

and immunofluorescent (IF) staining of Thbs4b localized adjacent to *mkxb* at the ch-SH MTJ (Fig. 3L-N,P-R).

The dermal bone tenocytes (cl-SH, cl-HYP, op-DO; cluster 30) subclustered into two populations that primarily differed in expression of MTJ (*thbs4b*) and enthesis (*klf2a*) markers (Fig. 4A,H,T). The 'enthesis' subcluster showed high *ifitm5* and *klf2a* expression, while the 'MTJ' subcluster expressed high levels of *thbs4b* (Fig. 4G). Unlike the ch-SH, *col22a1* expression was distributed across both 'enthesis' and 'MTJ' subclusters (Fig. 4G,T; Fig. S6B). *is*HCR combined with IF staining for mCherry in 72 hpf *Tg(scxa:mCherry)* embryos validated the subclustering, with *klf2a* expression specifically marking putative entheses of the cleithrum (cl-SH and cl-HYP) (Fig. 4B-F,I-M). *col22a1* and *thbs4b* expression similarly marked these entheseal cells as well as MTJ regions of the cl-SH and cl-HYP, suggesting that these genes, traditionally considered MTJ markers, can have variable expression in other tendon types (Fig. 4N-S,U-Y).

Three EOMT (cluster 19) subpopulations were identified through subclustering (Fig. 5A,G). Co-expression of *scxa* with the enthesis marker *klf2b* marked a putative ocular enthesis subcluster (Fig. 5G,F,L) and *col22a1* marked a putative ocular MTJ (Kult et al., 2021). The

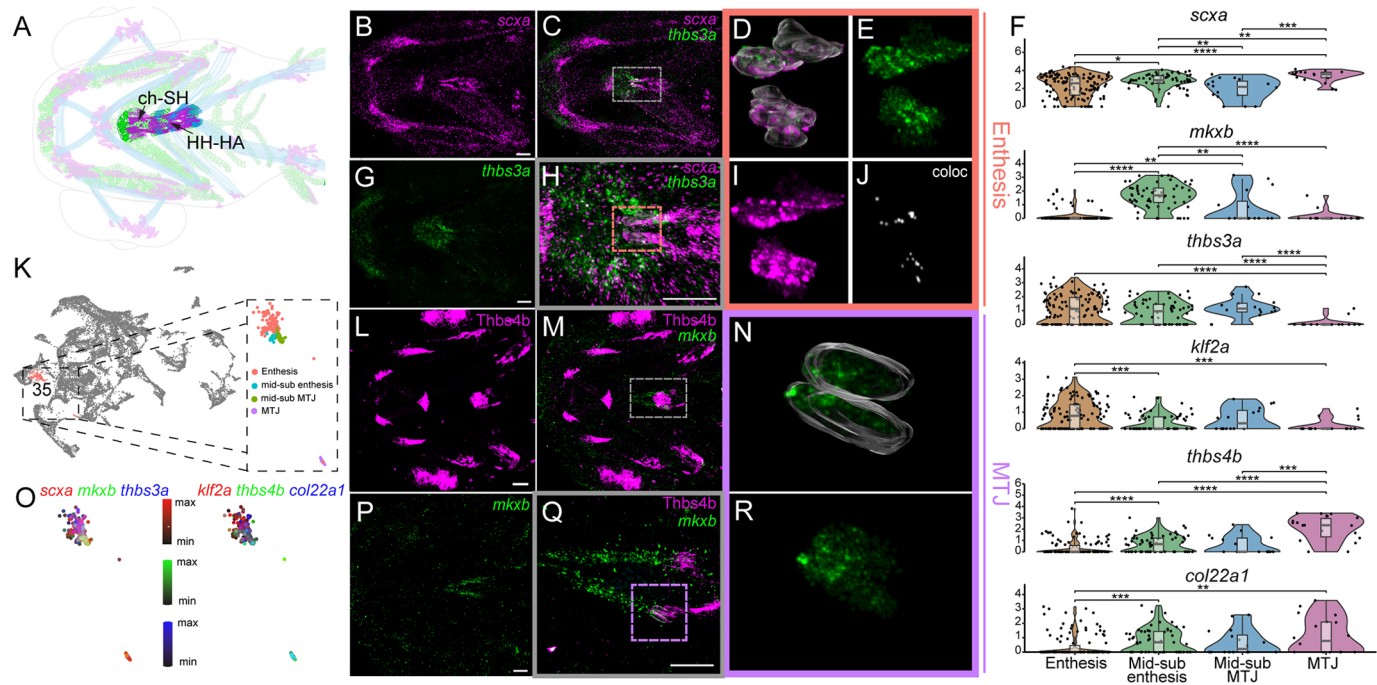

**Fig. 3. *is*HCR analysis confirms spatial heterogeneity within the ceratohyal-sternohyal tendons.** (A) Diagram of a ventral view of ch-SH in a 72 hpf *Tg(scxa:mCherry; sox10:eGFP)* embryo showing tenocytes (magenta), muscle fibers (blue) and chondrocytes (green). (B,C,G,L,M,P) Ventral views showing *is*HCR for *scxa* (B,C), *thbs3a* (C,G), *mkxb* (M,P) expression and immunofluorescence for Thbs4b (L,M). (H,Q) Magnified views of the region outlined by dotted lines in C and M, respectively. (D,E,I,J,N,R) Magnified views of the regions marked by dotted lines in H and Q showing traced 3D volumes of tenocyte nuclei containing expression of *scxa* (D,I), *thbs3a* (D,E), *scxa*/*thbs3a* colocalization (J) and *mkxb* (N,R). (F) Violin plots with overlayed bar plots displaying distributions of expression of tenocyte marker *scxa*, ch-SH tenocyte marker *mkxb*, ch-SH enthesis marker *thbs3a*, enthesis marker *klf2a* and MTJ markers *thbs4b* and *col22a1*, corresponding with clustering displayed in K. *P*-values calculated using Wilcoxon rank sum test. *$P<0.05$, **$P<0.01$, ***$P<0.001$, ****$P<0.0001$. The box region comprises 50% of cells, representing the interquartile range (IQR), while the bottom and top regions represent 25th and 75th percentile. The whiskers represent the spread of data outside the IQR within the normal variability of gene expression. (K) UMAP plot showing tenocyte subcluster 35 (ch-SH tenocytes) displaying ch-SH domains marked by expression differences in F and O. (O) Feature plot showing co-expression of *scxa*/*mkxb*/*thbs3a* corresponding to the location of the entheseal and mid-substance ch-SH population and *klf2a*/*thbs4b*/*col22a1* corresponding to the location of the MTJ and mid-substance MTJ ch-SH populations. Scale bars: 30 μm (B,C,G,L,M,P); 10 μm (H,Q).

third subcluster showed little *scxa*, *col22a1* and *klf2b* expression, suggesting a non-tenocyte periocular mesenchyme identity (Fig. S6C). *is*HCRs for *klf2b* or *col22a1* combined with IF staining for mCherry in 72 hpf *Tg(scxa:mCherry)* embryos labeled entheseal tenocytes within the scleral attachment of the inferior rectus (IR) muscle or putative MTJ adjacent to IR, respectively (Fig. 5B-E,H-K). Taken together, our data revealed distinct transcriptional signatures for entheseal and MTJ tenocytes that correlated with their hard or soft tissue tendon attachment surfaces across tendon types.

## Distinct tenocyte ECM expression between and within tendons and ligaments

To gain insights into functional differences between distinct tendons and ligaments, we examined ECM-specific gene expression of tissue-resident tenocytes/ligamentocytes. ECM-related genes were organized into categories that were used to compute module scores (Table 2) based on published studies (Charvet et al., 2013; Eisner et al., 2022; Frara et al., 2018; Halper, 2021; Ito et al., 2010; Jacobson et al., 2020; Jones et al., 2006; Juneja and Veillette, 2013; Karlsen et al., 2022; Maeda et al., 2010; Rosini et al., 2018; Savadipour et al., 2023; Shukunami et al., 2018; Van Der Rest and Garrone, 1991; Viehöfer et al., 2015; Yoon and Halper, 2005; Zhang and Wang, 2013). For module score comparisons across tissue types, clusters 19 (EOMTs), 29 (HH-HH), 16 (pec fin tendons), 35 (ch-SH) and 30 (dermal bone tendons) were combined to form a tendon cluster, clusters 12 (mc-mc) and 36 (mc-pq, pq-hs) were combined to

form a ligament cluster, and cluster 32 (mc-pq, mc-mc, ch-hs) comprised a joint cluster (with associated tenocytes) (Fig. 6A). Compared to tenocytes, ligamentocytes more strongly expressed tendon fibril collagen (TFC), tendon FACIT collagen (TFACIT), beaded filament collagen (BFC) and tendon matrix metalloproteinase (TMMP) while expressing lower levels of basement membrane collagen (BMC), microfibril elastin (MFE), lysyl oxidase (LO) and proprioception (Prp)-associated genes (Fig. 6A; Table 2). Such differences in collagen composition and collagen modulating factors suggest an important role for tailored ECM expression at the transcriptional level distinguishing tenocytes from ligamentocytes.

We also observed distinct ECM expression profiles between tenocytes of enthesis (grouped pec fin, dermal bone, ch-SH and EOMT enthesis subclusters) versus MTJ (grouped pec fin, dermal bone, ch-SH, EOMT and HH-HH MTJ subclusters) (Fig. 6B). Though module scores for all categories were significantly different in enthesis versus MTJ, the most striking differences were reduced TFC, BMC and BFC levels, and increased TFACIT expression in the enthesis (Fig. 6B). When we separated individual tendons of softer versus harder tissue attachments, we noticed a nuanced trend of TFC and TFACIT levels (Fig. 6C). For example, among entheseal tenocytes, we observed the highest TFC scores in ch-SH and dermal bone (stiffer attachments) tendons, and the lowest score in the EOMT tenocytes (softer attachments). Conversely, EOMT tenocytes had the highest TFACIT, BMC, tendon laminin (TL) and MFE scores, while tenocytes of the ch-SH and dermal bone had the lowest scores for these modules

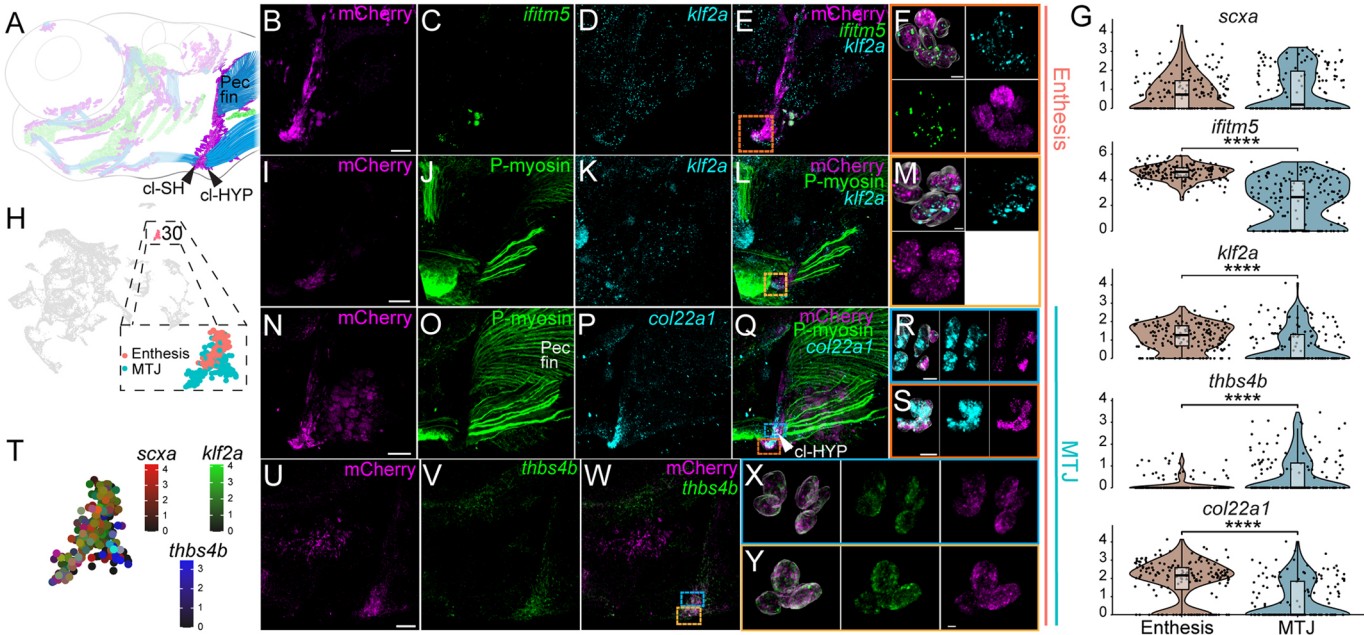

**Fig. 4. Subcluster and *is*HCR analyses of dermal bone and pectoral fin muscle attachments reveals spatial heterogeneity of MTJ and enthesis tenocytes.** (A) Diagram of lateral view of cranial musculoskeletal tissues in a 72 hpf embryo showing tenocytes (magenta), ligamentocytes (red), muscle fibers (blue) and chondrocytes (green) with cl-SH and pectoral fin tenocytes highlighted. (B-E,I-L,N-Q,U-W) Lateral views of 72 hpf embryos showing *is*HCR at the cl-SH tendon for *ifitm5* (C,E), *klf2a* (D,E,K,L), *col22a1* (P,Q) and *thbs4b* (V,W) and immunofluorescence for anti-mCherry (B,E,I,L,N,Q,U,W) and anti-phosphomyosin (P-myosin) (J,L,O,Q). (F,M) Magnified views of the region outlined by dotted lines in E and L, respectively, showing 3D-voxel volumes of tenocyte nuclei with mCherry staining, expression of *ifitm5* and *klf2a*. (G) Violin plots with overlayed box plots showing expression of *scxa*, *ifitm5*, *klf2a* and *col22a1* across the re-clustering of cluster 30. *P*-values calculated using Wilcoxon rank sum test. \*\*\*\**P*<0.0001. The box region comprises 50% of cells representing the interquartile range (IQR), while the bottom and top regions represent 25th and 75th percentile. The whiskers represent the spread of data outside the IQR within the normal variability of gene expression. (H) UMAP plot showing tenocyte cluster 30 (dermal bone tenocytes) re-clustered to display separation of distinct enthesis and MTJ domains marked by expression differences in G and T. (R,S,X,Y) Magnified views of the regions marked by blue and orange lines in Q and W displaying 3D voxel volumes of tenocyte nuclei with mCherry signal, and *is*HCR expression of *col22a1* and *thbs4b*. (T) Feature plot displaying relative expression distributions of *scxa*, *klf2a* (enthesis markers) and *thbs4b* (MTJ marker) across cluster 30. Scale bars: 30 μm (B,N,U); 2 μm (F); 20 μm (I); 3 μm (M); 5 μm (R,S,X,Y).

(Fig. 6C). Similar trends existed when comparing MTJ tenocytes, with increasing TFACIT, BMC, BFC, TL, TMMP and MFE module scores in dermal bone versus ch-SH versus EOMTs. The TFC, tendon proteoglycan (TP), tendon tissue inhibitors of metalloproteinases (TTIMP) and LO module scores showed a decreasing trend, with highest expression in the ch-SH MTJ, followed by the dermal bone, and the lowest in the EOMT. We observed a decreasing trend in Prp scores, highest in dermal bone MTJ tenocytes, followed by the ch-SH and the lowest in EOMT tenocytes, suggesting a correlation between force-responsiveness of tenocytes and size of attaching muscle (Fig. 6C) (Nayak et al., 2025). We also compared the ECM module scores between entheseal and MTJ tenocytes of pec-fin and ch-SH tendons, which, despite both being attached to cartilage structures at 72 hpf, have distinct developmental origins (ch-SH: cranial NC cells; pec-fin: lateral plate mesoderm). When comparing the entheseal tenocytes, the pec fin had higher TL and Prp module scores, whereas the ch-SH had higher TFC, BFC, TP, TTIMP and LO scores (Fig. 6C). When comparing the MTJ tenocytes of the pec fin and ch-SH however, the pec fin had higher BMC, BFC, TL, MFE and Prp scores, whereas the ch-SH had higher TFC, TFACIT, TMMP, TTIMP and LO scores. Finally, we compared ECM module scores between the MTJs of two types of soft tissue tendons – the EOMT versus the HH-HH, where the EOMT attaches to the eye sclera and the HH-HH is an intermuscular tendon. The EOMT MTJ tenocytes have higher TFACIT, TP, TMMP, TTIMP, MFE, LO and Prp scores, whereas the HH-HH MTJ tenocytes have higher BMC and BFC, with both attachments having equal scores in TFC and TL modules (Fig. 6C). These results suggest that ECM gene expression in tenocytes and

ligamentocytes vary based on their functional requirements and developmental origins.

## Spatially distinct tendons are characterized by unique tenocyte signaling environments

We have previously shown that tenocytes in zebrafish extend long projections into the ECM, likely acting as signaling sensors responding to changes in ECM microenvironment through TGFβ signaling (Subramanian et al., 2018). Similar processes have been described in mammals (Kalson et al., 2015; Knudsen et al., 2015; Subramanian et al., 2018), but despite characterization of select pathways like retinoic acid (RA) and TGFβ signaling, roles for other signaling pathways across spatially distinct tendons have not been reported (Kaji et al., 2021; McGurk et al., 2017). To address this, we performed KEGG pathway analysis on our scRNA-seq dataset using clusterProfiler to identify associations between relevant signaling pathways and individual clusters (Wu et al., 2021). The TGFβ pathway was significantly enriched in multiple clusters, including clusters 6 (pec fin tendons), 12 (mc-mc/iml), 19 (EOMTs), 29 (HH-HH) and 30 (dermal bone tendons), consistent with its role in tendon development and maintenance (Pryce et al., 2009; Subramanian et al., 2018) (Fig. 7A). Other enriched terms/pathways included Focal Adhesion (FAK) signaling and Wnt signaling. Wnt signaling was enriched in clusters 12 (mc-mc), 16 (pec fin tendons), 19 (EOMTs) and 29 (HH-HH). Although Wnt signaling has been implicated in migration and patterning of cranial NC cells, roles in tendon development have not been reported (Alexander et al., 2014; Devotta et al., 2018; Dranow et al., 2023;

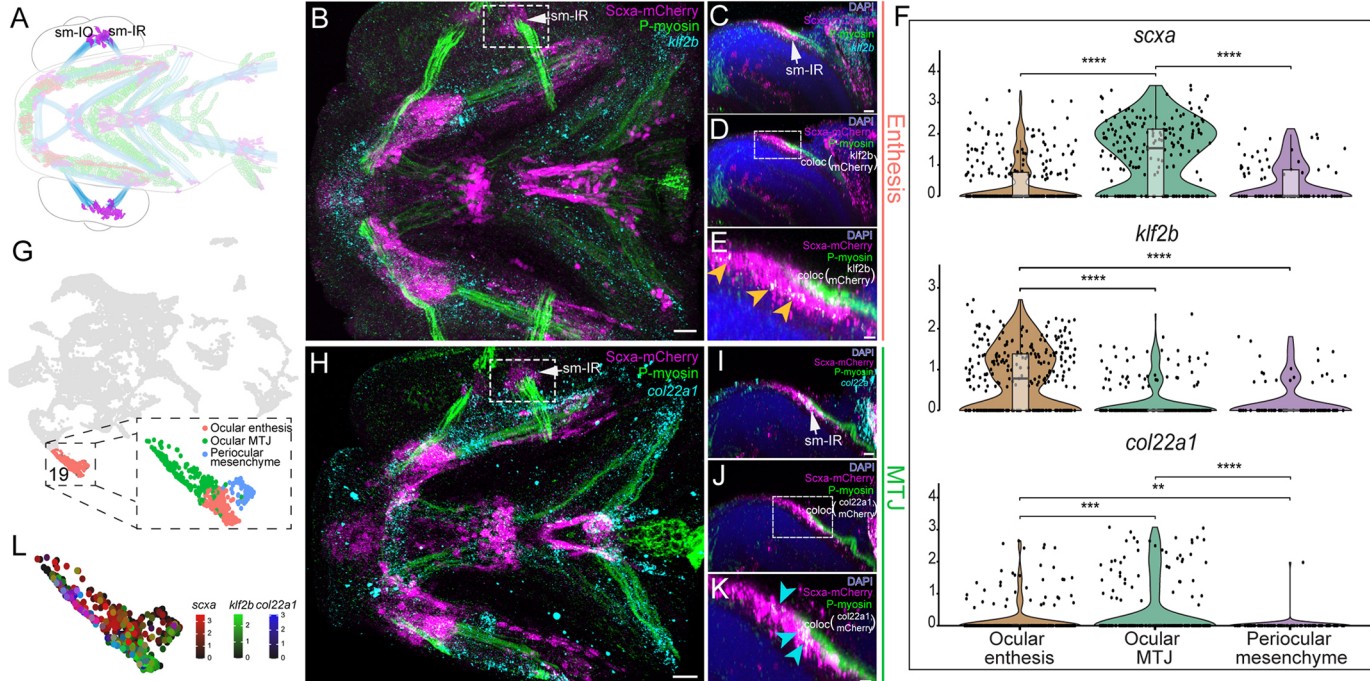

**Fig. 5. Subcluster and *is*HCR analyses showing spatial heterogeneity of extraocular tenocytes.** (A) Diagram of ventral view of the cranial musculoskeletal tissues in a *Tg(scxa:mCherry; sox10:eGFP)* 72 hpf embryo showing tenocytes (magenta), ligamentocytes (red), muscle fibers (blue) and chondrocytes (green) with the ventral extraocular muscle tendons (EOMTs) highlighted. (B,H) Ventral views of 72 hpf embryo heads stained with anti-mCherry and anti-Phosphomyosin (P-myosin) and *is*HCR for *klf2b* (B) and *col22a1* (H). (C-E) Oblique views of the region marked by dotted lines in B showing *is*HCR expression of *klf2b* (C), colocalization of anti-mCherry signal and *klf2b* (D) and a magnified view of the region marked by dotted lines in D, with orange arrowheads marking EOMT enthesis cells with colocalized anti-mCherry signal and *klf2b* expression (E). (F) Violin plots with overlayed box plots showing expression of *scxa*, *ifitm5*, *klf2a*, and *col22a1* across the re-clustering of cluster 30. *P*-values calculated using Wilcoxon rank sum test. **P<0.01, ***P<0.001, ****P<0.0001. The box region comprises 50% of cells representing the interquartile range (IQR), while the bottom and top regions represent 25th and 75th percentile. The whiskers represent the spread of data outside the IQR within the normal variability of gene expression. (G) UMAP plot displaying tenocyte cluster 19 (EOMT tenocytes) re-clustered to show separation of distinct ocular regions. (I-K) Oblique views of the region marked by dotted lines in H showing *is*HCR expression of *col22a1* (I), colocalization of anti-mCherry signal and *col22a1* (J) and a magnified view of the region marked by dotted lines in J with cyan arrowheads marking EOMT MTJ cells with colocalization of anti-mCherry signal and *col22a1* expression (K). (L) Feature plot of cluster 19 showing relative expression of *scxa*, *klf2b* and *col22a1*. Scale bars: 20 µm (B,H); 7 µm (C); 5 µm (E,I); 4 µm (K).

Mayor and Theveneau, 2014). We observed expression of Wnt receptors (such as *fzd1*, *fzd4*, *fzd5*) and downstream effectors such as *axin2* across different spatially distinct tendon regions, suggesting that tenocytes inhabiting different tendons have varied levels of Wnt activation (Fig. 7B,C).

To verify if cranial tenocytes are Wnt-responsive, we generated *Tg(scxa:mCherry;7Xtcf:gfp)* double transgenics and observed tenocytes co-expressing mCherry and GFP. At 48 hpf, we observed a small subset of mCherry⁺ tenocytes co-expressing GFP amidst GFP⁺ cells (representing canonical Wnt signaling) that formed two clusters on either side of the midline in the ventral mandibular arch and further posteriorly in the ventral hyoid arch (Fig. 8A-E; Movie 2). By 53 hpf, these midline clusters of co-expressing cells (populating the ventral jaw tendons at later stages) expanded (Fig. 8F-J). At 72 hpf, *is*HCR for *sox9a* (labeling chondrocytes) and immunostaining for Myosin Heavy Chain (MHC, labeling muscles) in *Tg(scxa: mCherry;7Xtcf:gfp)* embryos revealed that tenocytes/ligamentocytes exhibiting high Wnt signaling responsiveness were largely localized to the developing mc-mc ligament/joint region, mc-pq, pq-hs ligaments, HH-HH tendon, as predicted by our KEGG pathway analysis, and entheses of the ch-IH and ch-HH tendons (Figs 7A and 8K-X). Time-lapse imaging during cranial tendon formation showed dynamic Wnt responses in these developing tenocyte subsets, which increased in intensity as additional cranial attachments migrated (Movie 2).

## Gain or loss of canonical Wnt signaling at onset of cranial muscle attachment disrupts tenocyte patterning

To test roles for canonical Wnt signaling in cranial tendon formation, we crossed a heat shock inducible dominant-negative TCF-GFP transgenic *Tg(hsp70l:dnTCF-GFP)* with *Tg(scxa: mCherry)*. The 48 hpf double-transgenic embryos were heat shocked at 39°C for 30 min, when embryonic pharyngeal muscles and tendons are still migrating, and imaged live between 48 and 60 hpf (Fig. S7A-L; Movie 3). Tenocytes/ligamentocytes associated with mc-mc of heat-shocked *Tg(scxa:mCherry; hsp70l: dnTCF-GFP)* embryos failed to fully aggregate, mc-pq and pq-hs ligaments were reduced in size, and HH, IH and IMA/IMP muscles exhibited ectopic muscle attachments (Fig. 9A,B,D,E). Heat shocks typically caused reductions in Meckel's cartilage size, consistent with the role of Wnt in D-V arch patterning (Alexander et al., 2014). To confirm that muscle attachment defects were due to disruption of Wnt signaling, we crossed a heat shock inducible Wnt antagonist line, *Tg(hsp70l:Dkk1b-GFP)*, with *Tg(scxa: mCherry)* (Stoick-Cooper et al., 2007). Identical heat shock application at 48 hpf caused similar defects in condensations of ventral mc- and ch-associated tenocytes, though less severe than heat-shocked *Tg(scxa:mCherry; hsp70l: dnTCF-GFP)* embryos (Fig. 9A, B,F-I). Immunostaining of heat-shocked 72 hpf *Tg(scxa:mCherry; hsp70l:dnTCF-GFP)* and *Tg(scxa:mCherry; hsp70l:dkk1b-GFP)* embryos revealed ectopic muscle attachments associated with

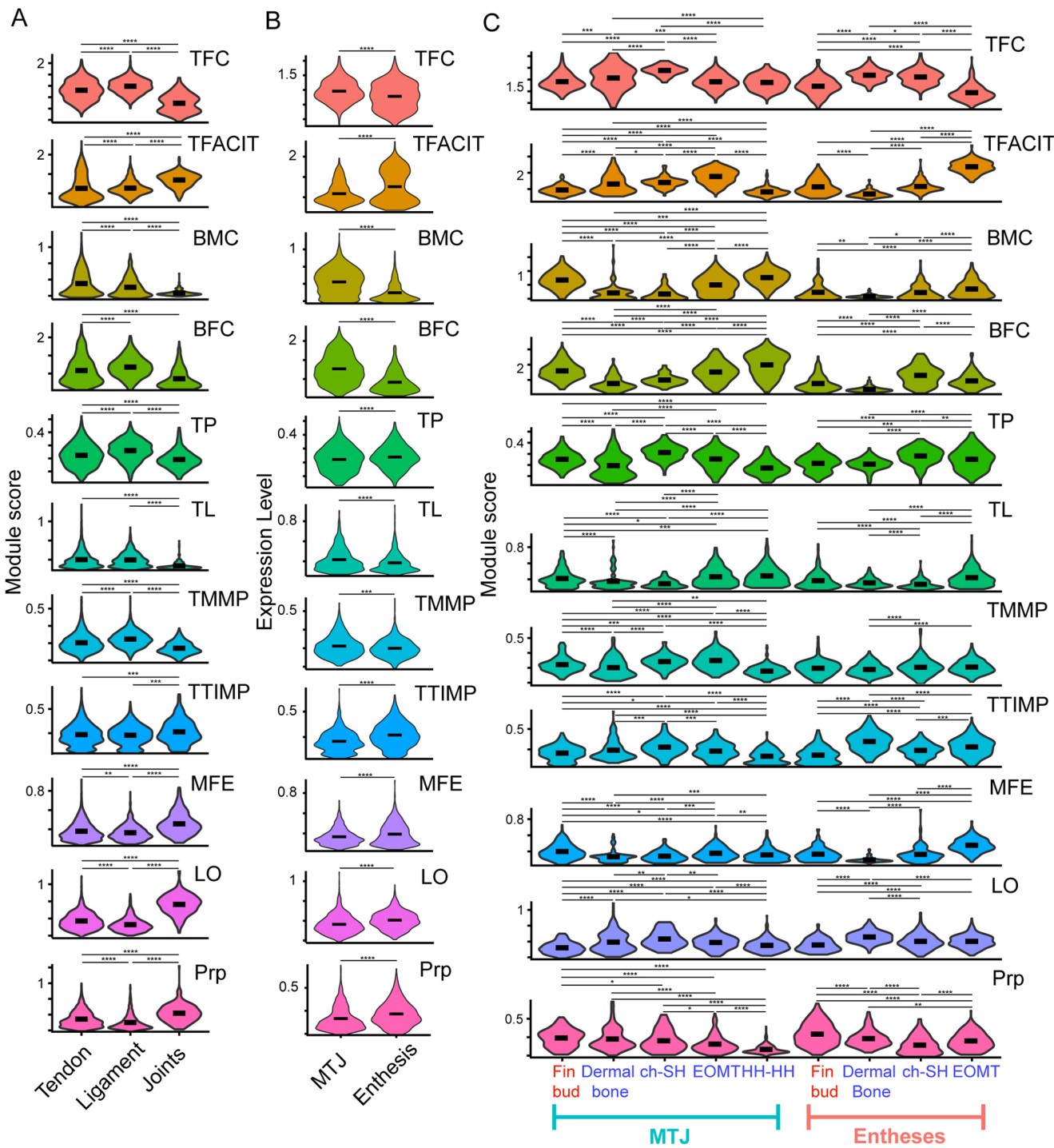

**Fig. 6. Spatially and functionally distinct tenocyte subclusters exhibit unique ECM transcriptional profiles.** (A) Violin plots showing relative module expression scores for various ECM- and proprioception-related genes between subclusters of tenocytes grouped as 'Tendons', Ligaments' and 'Joints' on the basis of isHCR-validated expression of marker genes. Groups are defined as: 'Tendons' includes clusters 19 (EOMTs), 29 (HH-HH), 16 (Pec Fin), 35 (ch-SH), and 30 (dermal bone tendons); 'Ligaments' includes 12 (iml/mc-mc) and 36 (mc-pq, pq-hs); 'Joints' includes cluster 32 (joint associated tenocytes). (B) Violin plots depicting various ECM- and proprioception-related module scores between tenocyte subclusters marking MTJ and enthesis of tendons associated with dermal bones, intermuscular attachments, EOM attachments, pectoral fins and cartilage. Groups are defined as: 'MTJ' includes Pec fin MTJ, EOMT MTJ, Dermal bone MTJ, ch-SH and HH-HH MTJ; 'Enthesis' includes Pec Fin enthesis, Dermal bone enthesis, ch-SH enthesis and EOMT enthesis. (C) Violin plots depicting various ECM- and proprioception-related module scores between individual tenocyte subclusters marking MTJ and enthesis of tendons associated with dermal bones, intermuscular attachments (HH-HH), EOM attachments, pectoral fins (Fin bud) and cartilage (ch-SH). Fin bud cluster gene expression (marked in red) was analyzed from single cell expression, while the gene expression differences in other tendons (marked in blue) were also supported by HCR-*in situ* expression experiments. Modules (see Table S1): tendon fibril collagen (TFC), tendon FACIT collagen (TFACIT), basement membrane collagen (BMC), beaded filament collagen (BFC), tendon proteoglycan (TP), tendon laminin (TL), tendon matrix metalloproteinase (TMMP), microfibril elastin (MFE), lysyl oxidase (LO), enthesis (E), myotendinous junction (MTJ), proprioception (Prp). All *P*-values calculated using Wilcoxon rank sum test. \*$P<0.05$, \*\*$P<0.01$, \*\*\*$P<0.001$, \*\*\*\*$P<0.0001$.

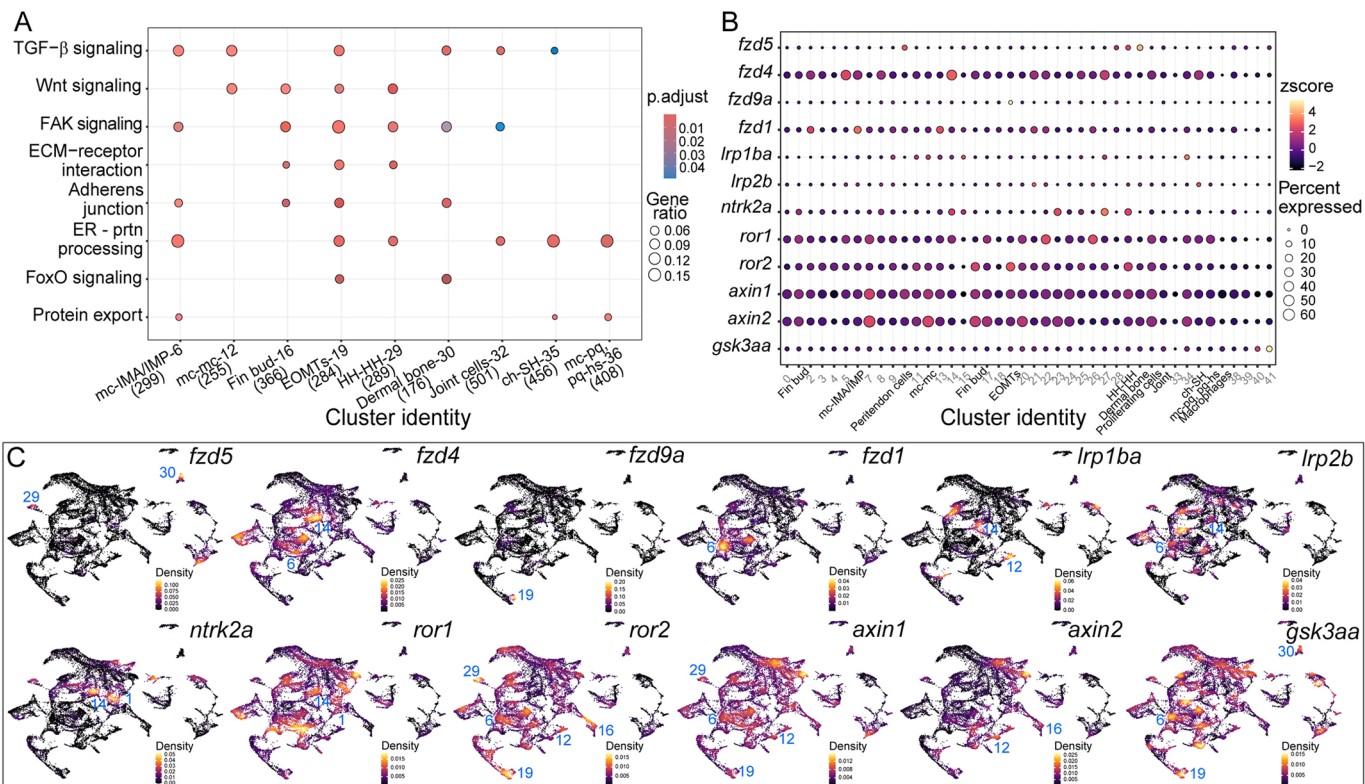

**Fig. 7. KEGG pathway analysis reveals unique signaling dynamics across tendon clusters.** (A) Dot plot of KEGG pathway analysis depicting top pathways mapped to *is*HCR-validated clusters within the scRNA-seq dataset. (B) Dot plot depicting expression of Wnt signaling-related genes across all clusters. (C) Nebulosa plots depicting distribution of expression of Wnt signaling-related genes across the scRNA-seq dataset.

scattered *scxa*:mCherry⁺ tenocytes or to other tendons (Fig. 9C; Fig. S7). Myofibers in heat-shocked embryos often appeared to be frayed. Heat shocks of 60 hpf *Tg(scxa:mCherry; hsp70l:dkk1b-GFP)* embryos followed by immunostaining at 72 hpf showed similar results (Fig. 9J-R), suggesting that Wnt signaling is required for establishment and maintenance of proper muscle-tendon attachment (i.e. 48-60 hpf). To determine if ectopic tenocyte-muscle attachments show ECM signatures of MTJs, we also heat shocked *Tg(scxa:mCherry; hsp70l: dkk1b-GFP)* embryos at 48 hpf and antibody stained with the MTJ marker Thbs4b. Thbs4b localized to an ectopic attachment, suggesting that ectopic muscle attachments are MTJs (Fig. S8).

To independently verify the role of Wnt in tendon development, we treated zebrafish embryos with two canonical Wnt antagonists, XAV939 and IWR-1 (Meyers et al., 2012; Yin et al., 2011), which reduce expression of downstream Wnt effector *axin2* in treated embryos (Wang et al., 2022). *Tg(scxa:mCherry;7Xtcf:gfp)* embryos treated with the Wnt antagonists from 48 to 72 hpf showed reduced GFP expression compared to DMSO-treated controls and exhibited defects in tenocyte condensation (Fig. 10A-Q; Fig. S9A-L) (Westphal et al., 2022). Many also displayed ectopic muscle attachments (Fig. 10J,Q; Fig. S9F,I,L). Conversely, we treated embryos with the Wnt agonist BIO to study the effect of upregulation of Wnt signaling on tenocyte patterning. We observed global increases in GFP expression in a dose-dependent manner in the developing head accompanied by ectopic muscle attachments, suggesting that precise modulation of Wnt signaling is crucial for proper MTJ formation (Fig. 10A-C,E,P-T; Fig. S9M-R).

## DISCUSSION

We have shown transcriptional heterogeneity among cranial tenocytes and ligamentocytes from comparative scRNA-seq of

*scxa*-expressing cells isolated from embryonic zebrafish. Cranial tenocytes segregate transcriptionally at multiple levels: (1) expression of patterning genes that specify dorsal, ventral and intermediate positional identities in the developing pharyngeal arches (Fig. S2); (2) expression of genes that correlate with the type of connective tissue (i.e. tendon, ligament or joint) and spatially distinct tendon (i.e. EOMT, dermal bone tendon, ch-SH); (3) expression of ECM genes that correlate with the attachment interface (i.e. enthesis or MTJ of spatially distinct tendons) (Askary et al., 2017; Yelick and Schilling, 2002). This suggests that tenocytes in an individual tendon attachment zone are transcriptionally tuned to suit their attachment interface. While our data from zebrafish cranial tenocytes share similarities with scRNA-seq studies using mammalian Achilles tendons as well as enthesis and MTJ samples mainly from limbs of mice and humans, ours is the first to profile tenocytes across different tendons during embryonic development and establish spatial and functional ECM expression signatures (Karlsen et al., 2022; Kult et al., 2021; Petrany et al., 2020; Steffen et al., 2023; Tan et al., 2020; Zhang et al., 2023). We have also uncovered a functional role for Wnt signaling in the development and patterning of cranial MTJs. Our results reveal key transcriptional differences both within and across spatially distinct tendons during their embryonic development.

## Spatial tendon transcriptional signatures correlate with attachment strength and function

Higher order clustering revealed spatially distinct tenocytes co-expressing unique markers with *scxa* (e.g. *mkxb* – SHT tenocytes; *fndc1* – ligamentocytes) that distinguish tendons from one another and that correlate with expression of selected ECM marker genes. We verified co-expression using *is*HCR *in vivo* to annotate distinct tendon, ligament and joint clusters, as well as spatially distinct

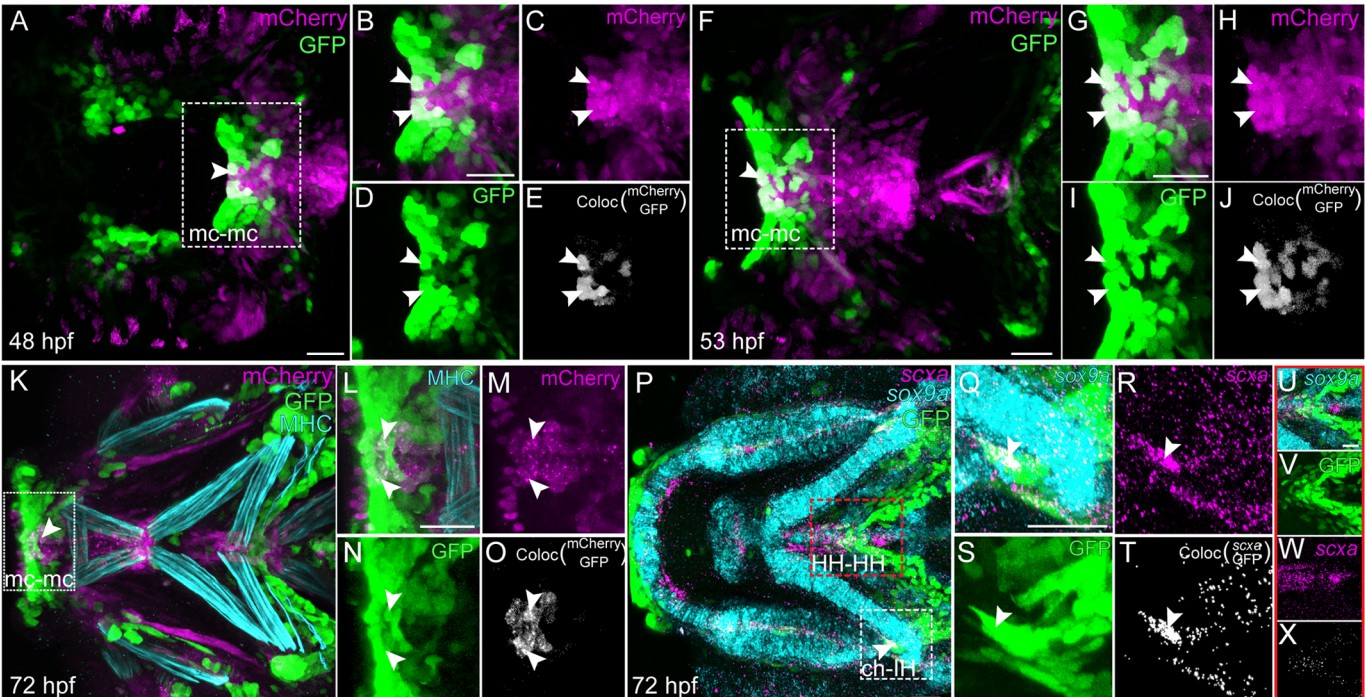

**Fig. 8. Subsets of tenocytes are marked by active Wnt signaling during development.** (A,F,K) Ventral views of cranial tendons at 48 hpf (A), 53 hpf (F) and 72 hpf (K) in *Tg(scxa:mCherry; 7xTCF:GFP)* zebrafish embryos immunostained for anti-mCherry (tenocytes, magenta), anti-GFP (Wnt responsive cells, green) and anti-MHC (muscle fibers, blue). (P) Ventral view of *is*HCR-stained *Tg(7xTCF:GFP)* 72 hpf embryo showing expression of *scxa* in tenocytes (magenta), *sox9a* in cartilage (blue) and co-immunostained with anti-GFP (Wnt responsive, green). (B-D,G-I,L-N,Q-S) Magnified views of region marked by white dotted lines in A,F,K,P, respectively, showing immunofluorescent signal of mCherry and GFP and *is*HCR expression of *scxa*, *sox9a* and IF staining of GFP. (E,J,O,T) Monochrome panel showing tenocytes co-expressing mCherry and GFP, and *scxa* and GFP. (U-W) Magnified view of region marked by red dotted lines in P showing *is*HCR expression of *scxa*, *sox9a* and IF staining of GFP. (X) Monochrome image showing tenocytes co-expressing *scxa* and GFP. Scale bars: 20 μm (A,F,K,Q,U); 30 μm (P); 15 μm (G); 10 μm (B,L).

tendons. Gene modules based on published tendon gene expression, proteomics and our own *is*HCR data, also distinguish transcriptional programs regulating ECM gene expression associated with regions within tendons, such as MTJ, mid-substance and enthesis subdomains, (Tai et al., 2017; Ideo et al., 2020; Kaku et al., 2024; Karlsen et al., 2022; Kult et al., 2021; Yan et al., 2022). Further sub clustering also reveals differences between tenocytes associated with skeletal tissues such as cartilage and bone or softer tissues such as the scleral surface of the eye, some of which correlate with tissue stiffness.

Ligament and tendon ECMs are both rich in fibrillar collagens (e.g. COL1, COL3) though ligament fibril bundles are less parallel and higher in elastin microfibrils involved in cross-linking (Tai et al., 2017; Trębacz and Barzycka, 2023). This confers a nonlinear, anisotropic property to the ligament ECM, with increasing stiffness as loads increase, which limits articulation and stabilizes joints (Frank, 2004). Our data reveal several markers of joint, mc-pq and pq-hs ligamentocytes likely essential for the function of ligament ECM (Trębacz and Barzycka, 2023) such as *emilin3a*, the biological role of which in ligaments is not well understood. While we observe *emilin3a* expression at 72 hpf in cranial joint chondrocytes and associated tenocytes, previous studies have shown *emilin3a* expression in the developing notochord and floor plate of zebrafish, as well as the tail bud, primitive gut, gonads and osteogenic mesenchyme in mice, with a possible role in the regulation of TGFβ ligand activation (Milanetto et al., 2008; Schiavinato et al., 2012). In zebrafish, TGFβ-mediated mechanotransduction regulates tenocyte transcriptional programs in entheses and at somite boundaries in zebrafish (Subramanian et al., 2018, 2023). Similar to *emilin3a*,

*fndc1*, encoding a myokine that binds to α5β1 integrin receptors and is likely secreted in response to muscle contraction or injury, is also expressed in cranial joint cells and in the mc-pq-hs ligament (Zhang et al., 2024).

We further characterized new tendon markers, such as *angptl7* and *thbs5*. Interestingly, *angptl7* marks most cartilage-associated cranial tendons as well as EOMTs. While the function of Angptl7 in tendons is unknown, a close family member, Angptl4, promotes mammalian tenocyte proliferation, adhesion and migration (Gur-Cohen et al., 2019; Jamil et al., 2017). Similar to Thbs4, tenocytes also express Thbs5 at developing MTJs, where it may play roles in injury repair and it is also associated with pseudoachondroplasia, a condition in which chondrocyte proliferation and secretion of fibrillar collagens associated with tendons/ligaments are affected (Hecht et al., 1998; Posey et al., 2018). Thbs5 interacts with Aggrecan, TGFβ and fibrillar collagens to regulate tendon and articular cartilage ECM organization and repair (Chen et al., 2007; Gebauer et al., 2018). Future functional studies will reveal the functional roles of these proteins in joint and ligament development and maturation.

### Distinct ECM expression patterns differentiate tendons, ligaments and joints and mark intra-tendon functional domains

Previously published studies have isolated MTJ and entheseal tissue from limb tendons to profile tenocyte transcriptional signatures at these interfaces (Karlsen et al., 2022; Kult et al., 2021; Malbouyres et al., 2022; Yan et al., 2022). Here, we profile enthesis and MTJ tenocytes across different types of embryonic cranial tendons (e.g.

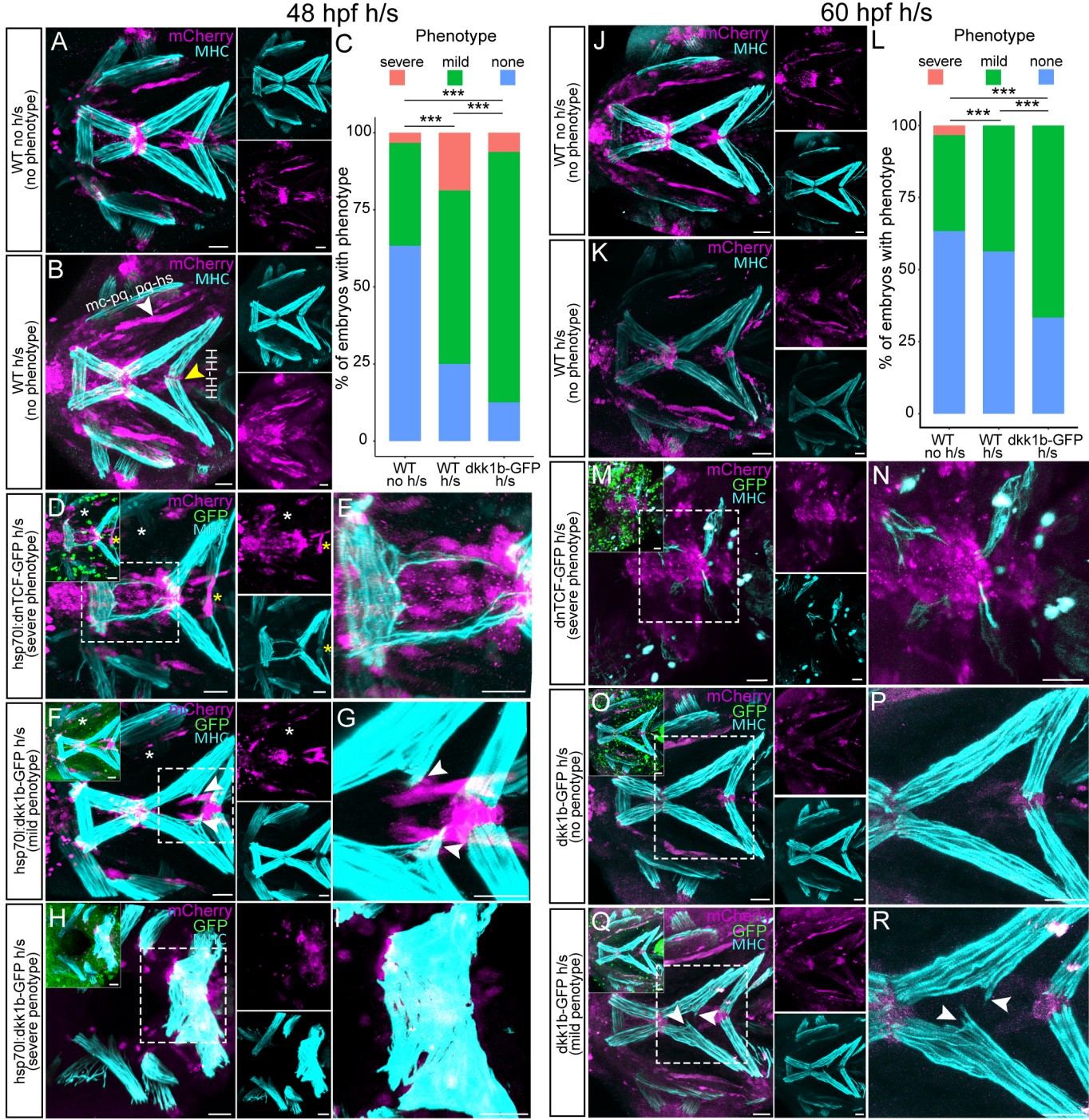

**Fig. 9. Genetic perturbation of Wnt signaling disrupts muscle attachments and tendon pattern.** (A-R) Ventral views of cranial tendons at 48 hpf and 60 hpf *Tg(scxa:mCherry)*, *Tg(scxa:mCherry; hsp70l:dkk1b-GFP)* and *Tg(scxa:mCherry; hsp70l:dnTCF-GFP)* embryos that were heat shocked and immunostained for anti-mCherry (tenocytes), anti-GFP (Wnt responsive cells) and anti-MHC (muscle fibers). Images show muscle attachments in untreated (A,J) and heat-shocked controls (B,K), *Tg(scxa:mCherry)* and heat-shocked transgenics with perturbed Wnt signaling (D-I, M-R). (C,L) Histograms show distribution of normal, mild and severe muscle attachment defects between control and heat-shocked *Tg(scxa:mCherry; hsp70l:dkk1b-GFP)* embryos. *P*-values were calculated using chi-square test of independence. \*\*\**P*<0.001 (48 hpf, *n*≈79; 60 hpf, *n*≈76). Scale bars: 30 μm.

skeletal, soft tissue attachments) using established markers and identify new expression patterns (*klf2b*, *thbs3a*, *klf2a*) from differentially expressed gene analysis of functionally distinct tendon types (EOMT and ch-SH versus dermal bone). The results support a model in which entheseal tenocytes have unique transcriptional signatures that correlate with the stiffness of the attachment surface. We have previously shown that *klf2a* expression

at a cartilage enthesis is developmentally regulated by the force of muscle contraction (Nayak et al., 2025). In contrast, MTJ tenocytes appear to share transcriptional signatures that lack *sox9a* and *klf2a* and include *thbs4b* and *col22a1*, as previously described (Charvet et al., 2013; Subramanian and Schilling, 2014).

Having validated clusters corresponding to tissue-specific tenocytes/ligamentocytes (tendons, ligaments, joint-adjacent tissue)

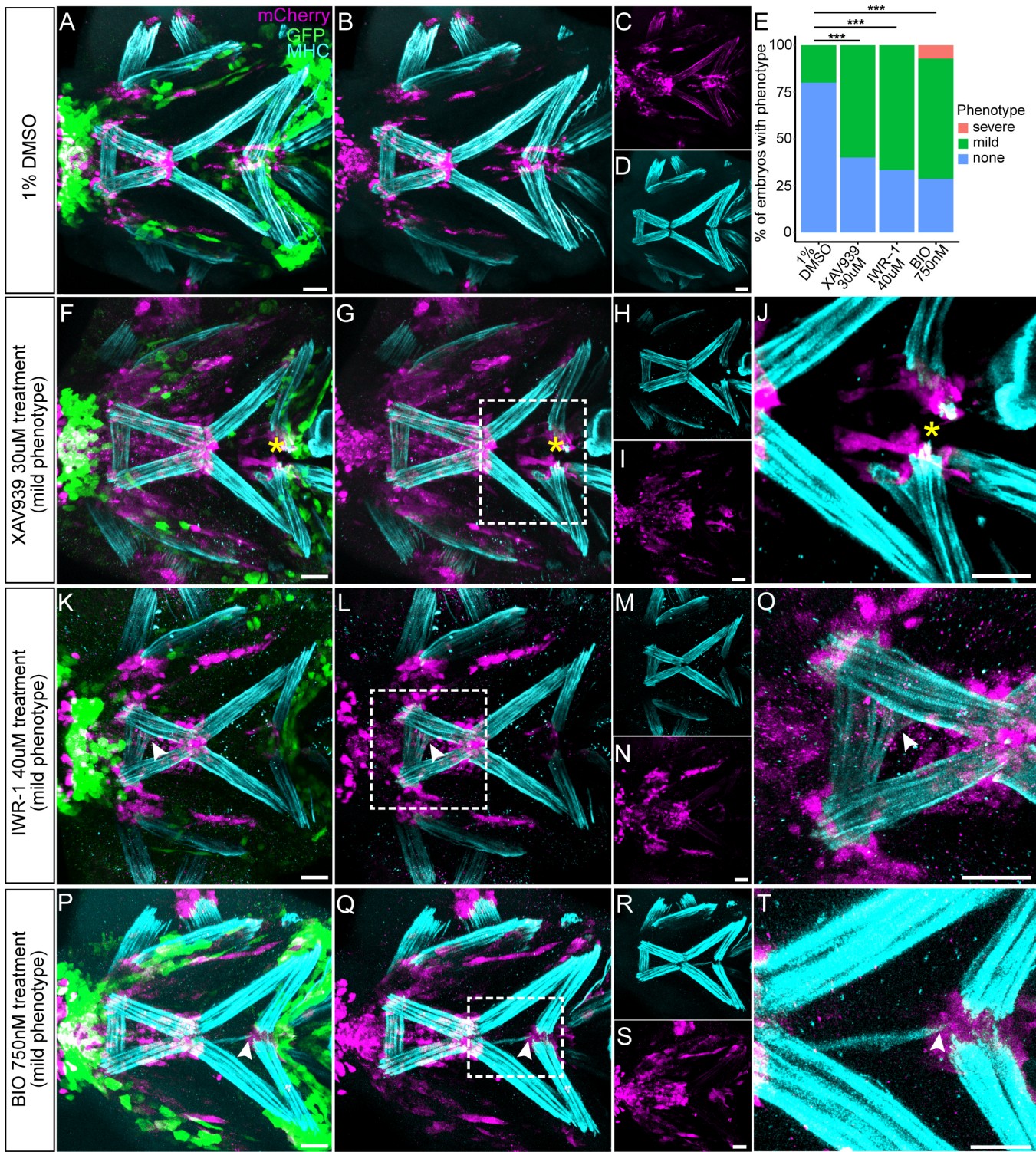

**Fig. 10. Pharmacological perturbation of Wnt signaling causes ectopic muscle attachments and defects in tenocyte aggregation.** (A-T) Ventral views of cranial tendons at 72 hpf *Tg(scxa:mCherry;7xTCF:GFP)* embryos that were treated with DMSO (control) (A-D), Wnt antagonist XAV939 (F-J), Wnt antagonist IWR-1 (K-O) and Wnt agonist BIO (P-T) immunostained for mCherry, myosin heavy chain (MHC) and GFP. (J,O,T) Magnified views of region marked by dotted lines in G,L,Q, respectively, showing ectopic muscle attachments to tenocytes. (E) Histogram displaying quantification of distribution of normal, mild and severe muscle attachment phenotypes between control and treated embryos. *P*-values were calculated using chi-square test of independence. ***P<0.001 (*n*≈60). Scale bars: 30 μm.

we propose that the underlying functional differences between these tissue types correlate with their ECM transcriptional signature. For example, we observe highest expression of fibrillar collagens, beaded-filament (collagen VI), proteoglycans and matrix

metalloproteinases in ligaments. This correlates with the high load and structural requirements of ligaments compared to tendons that are necessary for regulating joint range-of-motion guiding and stabilization (Benjamin et al., 2008; Cescon et al., 2015; Frank,

2004). Joint-adjacent tenocytes express the highest levels of FACIT collagens, genes associated with elastin fibrillogenesis (MFE), LO, and TTIMP, which suggest active fibrillogenesis and complement the functional requirements of joints (Marturano et al., 2014; Van Doren, 2015).

Published studies show a transition at the MTJ from a fibrillar tendon architecture to the basement membrane ECM composed of FACIT collagens, BMC and laminins (Jacobson et al., 2020; Jenkins et al., 2016; Malbouyres et al., 2022). This is reflected in our module comparisons where TFC, TFACIT, BMC, BFC, TL and TMMP are higher in tenocytes at the MTJ compared to those at the enthesis. Entheseal tenocytes also show low levels of BMC and BFC expression. Col IV (BM collagen) forms specialized sulfilimine bonds within basement membranes to increase mechanical stress resistance, and Col VI (BF collagen) orchestrates formation of crucial cross-links with fibrillar collagen bundles from tendons to insert and form aponeuroses and support the transfer of mechanical force (Benjamin et al., 2008; Cescon et al., 2015; Vanacore et al., 2009). Compared to the MTJ, entheseal tenocytes show higher expression of MFE, LO and Prp genes, suggesting reorganization and fibrillogenesis at the enthesis associated with mechanotransduction, consistent with the force-dependent regulation of *scxa* to *sox9a* ratio in entheseal tenocytes (Subramanian et al., 2023).

Tendons form attachments with interfaces of varying stiffness. ECM module expression scores show unique expression patterns between entheses that differ in stiffness of attachment (fin bud – cartilage, dermal bone, ch-SH – cartilage, EOMT – sclera). For example, TFC, LO and TMMP modules that associated with fibrocartilage matrix show highest expression in the dermal bone tenocyte cluster. One hypothesis is that the cleithrum (dermal bone), attached to both the large hypaxial muscle (cl-HYP) at its posterior end and the sternohyoideus (cl-SH) at its anterior end, experiences relatively large and continuous force due to swimming and suction feeding, thus requiring the mechanical strength and flexibility conferred by a high proportion of fibrillar collagens relative to other collagen types (Diogo et al., 2008; Hernandez, 2002; Schilling and Kimmel, 1997; Van Meer et al., 2025). EOMT (soft tissue entheses) are associated with scleral tissue and function in positioning the eye and are likely exposed to much weaker forces. EOMT show highest expression of BMC, TL, MFE and TFACIT modules associated with basement membrane ECM organization and structural integrity. Elastins and FACIT collagens (e.g. Col XII) may have roles in preserving tissue elasticity and extensibility (Fung et al., 2022; Ishizaki et al., 2024). Fin bud entheses, which are patterned from lateral plate mesoderm, show expression profiles similar to ch-SH entheses (NC-cell derived), but with slight variations, suggesting influences from both developmental cues and muscle function.

Tenocytes at the MTJ also show unique ECM expression profiles across spatially distinct tendons. Basement membrane-related ECM genes, such as BMC, BFC and TL modules, show highest expression in soft tissue tendon MTJ (EOMT), intermuscular attachment (HH-HH) and fin bud MTJ (Ishizaki et al., 2024; Silver et al., 2002). MTJs associated with stiffer tendons (dermal bone and ch-SH) are associated with higher expression of TFC, LO and TTIMP modules. Dermal bone and ch-SH MTJs are associated with transfer of larger magnitudes of force requiring stronger ECM matrix. These results suggest that transcriptional differences across tenocytes depend on the dynamics of force distribution across tendon tissue, developmental patterning cues and properties of the underlying surface tissue attachment.

## Wnt signaling promotes tenocyte aggregation in spatially distinct cranial tendons

Wnt signaling in tenocytes has been linked to downregulation of expression of traditional tenocyte markers such as *Scx* and *Mkx* (García-Lee et al., 2021; Kishimoto et al., 2017). In our imaging of *Tg(scxa:mCherry;7XTCF:GFP)* embryos, we observed restriction of Wnt-responsive tenocytes (GFP+/mCherry+) to the HH-HH, ch-IH, mc-pq-hs and mc-mc, suggesting a different role of Wnt signaling in these subpopulations from those of other cranial tendons. The results from our heat-shocked *Tg(scxa:mCherry; hsp70l:dkk1b-GFP)* and *Tg(scxa:mCherry; hsp70l:dnTCF-GFP)* perturbations show that tenocyte patterning depends on Wnt signaling at stages later than and distinct from the function of canonical Wnt signaling in migration of NC cells or non-canonical signaling in skeletal patterning (Alexander et al., 2014; Dorsky et al., 1998; Dranow et al., 2023; Tuttle et al., 2014). We postulate that the observed phenotype is due to loss of tenocyte progenitor cell adhesion, as ectopic tenocytes and higher frequencies of ectopic muscle attachments occur even in 60 hpf heat-shocked *Tg(scxa:mCherry; hsp70l:dkk1b-GFP)* embryos, a stage at which tenocytes have already migrated to their final locations. Interestingly, both wild-type (WT) 60 hpf heat-shocked and *Tg(scxa:mCherry; hsp70l:dkk1b-GFP)* 60 hpf heat-shocked embryos displayed no 'severe' phenotypes such as those seen in the *Tg(scxa:mCherry; hsp70l:dkk1b-GFP)* 48 hpf heat-shocked embryos, suggesting that the phenotype observed at 48 hpf is likely due to combined roles of Wnt signaling in global craniofacial patterning at this stage (Fig. 9). Small molecule antagonists of Wnt signaling caused similar ectopic muscle attachments as did the Wnt agonist (BIO), suggesting that an optimal level of Wnt signaling is required for proper patterning of cranial MTJs. Since all the Wnt disruptions (heat shock and drug treatments) were global, tissue-specific roles for Wnt signaling in tendon patterning remain unclear.

Collectively, our results establish a comprehensive map of transcriptionally distinct tenocyte clusters in spatially distinct cranial tendons and ligaments, which show unique profiles of ECM gene expression that correlate with their functions. In addition, they reveal new roles for canonical Wnt signaling in cranial tendon morphogenesis. Further studies of the genes identified by these single-cell studies will lead to a clearer understanding of the differentiation trajectories of tenocytes and improve therapies for human tendon/ligament injuries and tendinopathies.

## MATERIALS AND METHODS
### Zebrafish embryos, transgenics and mutants
AB strain WT (RRID:NCBITaxon_7955) and *TgBAC(scxa:mCherry)fb301* (RRID:ZFIN_ZDBGENO-180925-6), *Tg(7xTCF-Xla.Sia:GFP)ia4* (RRID: ZIRC_ZL7488), *Tg(hsp70l:dkk1b-GFP)w32* (RRID:ZIRC_ZL2050), *Tg(hsp70l:tcf7l1a-GFP)w26* (RRID:ZIRC_ZL509) [referred to in the text as *Tg(scxa:mCherry)*, *Tg(7XTCF:GFP)*, *Tg(hsp70l:dkk1b-GFP)* and *Tg(hsp70l:dnTCF)* respectively], were collected in natural matings, raised in embryo medium (EM) at 28.5°C (Westerfield, 2000) and staged as described (Kimmel et al., 1995).

### scRNA-seq sequencing
Dissociated cell suspensions from cold and warm conditions were sorted on a Bio-Rad FACS Aria Fusion cell sorter located at the UCI Institute for Immunology Flow Cytometry Facility (RRID:SCR_026616). Sorted mCherry+ cell suspensions were provided to the UCI Genomics High Throughput Facility (GHTF) (RRID:SCR_026615) for 10x library preparation using 3' v3 chemistry (10x Genomics) and sequenced.

### Processing of raw reads
FASTQ reads for all conditions obtained from GHTF were mapped to zebrafish genome version GRCz11 using CellRanger (version 3.1.0;

10x Genomics Cellranger DNA; RRID:SCR_023221) after modifying the genome and GTF annotations with the addition of the mCherry nucleotide sequence (708 nucleotides). Web summary metrics post-alignment for three samples were:

Sample 1 – estimated number of cells: 14,531; mean reads per cell: 37,963; median genes per cell: 1960; number of reads: 551,646,943; valid barcodes: 97.8%; valid UMIs: 100%; sequencing saturation: 54.2%; reads mapped to genome: 93.1%; reads mapped confidently to genome: 78.6%; reads mapped confidently to transcriptome: 61.9%.

Sample 2 – estimated number of cells: 11,109; mean reads per cell: 60,734; median genes per cell: 1786; number of reads: 674,692,117; valid barcodes: 97.3%; valid UMIs: 100%; sequencing saturation: 73.6%; reads mapped to genome: 93.6%; reads mapped confidently to genome: 85.6%; reads mapped confidently to transcriptome: 69.1%.

Sample 3 – estimated number of cells: 11,040; mean reads per cell: 93,951; median genes per cell: 2277; number of reads: 1,037,216,198; valid barcodes: 96.1%; valid UMIs: 100%; sequencing saturation: 79.0%; reads mapped to genome: 95.5%; reads mapped confidently to genome: 87.8%; reads mapped confidently to transcriptome: 66.0%.

### scRNA-seq quality control and analysis for 72 hpf heterogeneity study

Filtered count matrices for each condition were converted into Seurat objects (version 4.0.5, R version 4.0.2) (Seurat, RRID:SCR_016341) (Hao et al., 2021). Cells were kept for downstream analysis if they met the quality control criteria of 200>$n$Features>3000 and mitochondrial gene expression <5% for two samples and integrated with the cold protease condition sample described above. For anchoring/sample integration, individual Seurat objects were merged together using the 'merge' function and data were normalized using the 'NormalizeData()' function with default parameters. Feature selection was carried out using the 'FindVariableFeatures' function with default parameters. Data were scaled with the 'ScaleData()' function, principal component analysis (PCA) was performed using the 'RunPCA' function with $npcs=20$. All 20 PCs were used for UMAP reduction using 'RunUMAP()' and nearest neighbor graph construction using 'FindNeighbors()'. Unsupervised clustering was performed on the *is*HCR verified spatially distinct tendon UMAP (Fig. 2) with the 'FindClusters()' function using a resolution parameter of *resolution*=1. Doublets were removed using the DoubletFinder package (DoubletFinder, RRID:SCR_018771), with default settings. Module scores were calculated using the 'AddModuleScore()' function on the entire dataset with default settings. All statistics involving pairwise module score comparisons were performed with Wilcoxon rank sum test using 'stat_compare_means()' from the 'ggpubr' package (version 0.6.0) (ggpubr, RRID:SCR_021139). KEGG pathway analysis (KEGG enrichment analysis, RRID:SCR_027172) was performed using the 'enrichKEGG()' function in clusterProfiler (RRID:SCR_016884) (Wu et al., 2021), with default values for *P*-value, Bejamini-Hochberg test correction and q-value. Violin plots in Figs 1, 3, 4, 5 were generated using the SeuratExtend package (SeuratExtend, RRID:SCR_026143) (Hua et al., 2025).

### HCR and immunohistochemistry

*is*HCR probes were designed by Molecular Instruments and whole mount *is*HCR was performed using amplifiers/probes obtained from Molecular Instruments according to the *is*HCR v3.0 protocol as described in Choi et al. (2014). For each isHCR experiment, 10-12 embryos were used. Probes/amplifier combinations used are provided in Table S3.

Immunofluorescence staining was performed according to the protocols from Molecular Instruments and as previously described (Choi et al., 2010; Nayak et al., 2025). Embryos were stained using mouse monoclonal anti-Myosin heavy chain (MHC) antibody (Developmental Studies Hybridoma Bank, A4.1025; RRID:AB_528356) at 1:200 dilution, rat monoclonal anti-mCherry antibody (Invitrogen, M11217; RRID:AB_2536611) at 1:500 dilution, rabbit polyclonal anti p-Myosin light chain (p-MLC) antibody (Cell Signaling Technology, 3671; RRID:AB_330248) at 1:500 dilution, rabbit anti-Thrombospondin4b antibody (Thbs4b) (GeneTex, GTX125869; RRID:AB_2885605) at 1:1000 dilution. The following secondary antibodies from Jackson ImmunoResearch were used for fluorescence staining at 1:1000 dilution: AlexaFluor 594 AffiniPure donkey anti-rat IgG (712-586-

153; RRID:AB_2340691), Alexa Fluor 488-conjugated donkey anti-mouse IgG (715-546-150; RRID:AB_2340849), Alexa Fluor 647 conjugated AffiniPure donkey anti-rabbit IgG (711-606-152; RRID:AB_2340625), Alexa Fluor 594 AffiniPure donkey anti-rabbit IgG (711-586-152; RRID: AB_2340622).

### Imaging

Embryos were imaged (both images and timelapse footage) on a Leica SP8 (RRID:SCR_013673), and images were analyzed on IMARIS software (version 10.0.1) (RRID:SCR_007370) at the Optical Biology Core facility at UCI (RRID:SCR_026614). Colocalizations in images represent voxel colocalizations, measured through the IMARIS 'Coloc' function. Voxel colocalization shows overlap of fluorescent channels within a particular voxel which, in the case of *is*HCRs, may not reflect actual colocalization of fluorescence within a particular cell in some instances due to the punctate nature of *is*HCR fluorescence.

### Cold protease (Subtilisin A) embryo dissociation protocol and single-cell isolation for 10x sequencing

Embryo dissociation was performed as described previously (Subramanian et al., 2025). High expressing mCherry⁺ cells were gated and FACS sorted.

### Wnt signaling heat shock treatments

*Tg(scxa:mCherry)* fish were crossed with *Tg(hsp70l:dkk1b-GFP)* and *Tg(hsp70l:dnTCF)* lines to obtain *Tg(scxa:mCherry; hsp70l:dkk1b-GFP)* and *Tg(scxa:mCherry; hsp70l:dnTCF)* lines, respectively. Embryos were placed individually in 0.2 ml eight-strip PCR tubes (USA Scientific, 1402-2500) in 100 µl EM with an air-bubble placed at the bottom of each tube and heated at 39°C for 30 min in a thermocycler (Bio-Rad S1000). Then, 20 min after heat shock, GFP⁺ embryos were screened, fixed, antibody stained, mounted in 1% low melt agarose in PBS in a Glass Bottom Slide dish (MatTek Corporation, P35G-1.5-14-C) at 48 hpf and 60 hpf along with *Tg(scxa:mCherry)* and *Tg(scxa:mCherry)* heat-shocked controls at the same stages, and imaged.

### Wnt signaling drug treatments

*Tg(scxa:mCherry)* were crossed with *Tg(7XTCF:GFP)* to obtain *Tg(scxa:mCherry;7XTCF:GFP)* double positive embryos. 10 mM stocks of IWR-1 (Sigma # I0161, Pubchem CID: 44483163), XAV939 (Sigma # 575545, Pubchem CID: 135418940), and BIO (Sigma # B1686, Pubmed CID: 448949) were created by dissolving in DMSO (99.9+%, Alfa Aesar, #42780, Pubchem CID: 679). For experiments, 10 mM stocks were diluted in 3 ml EM to create working concentrations and added to ~15 embryos in a 35×10 mm petri dish (Falcon, #351008) per condition. Dishes with 48 hpf embryos were incubated at 28.5°C for 24 h, fixed, antibody/HCR stained, mounted in 1% low melt agarose in PBS in a Glass Bottom Slide dish (MatTek Corporation P35G-1.5-14-C) and imaged.

### Live imaging

For *Tg(scxa:mCherry)*, *Tg(scxa:mCherry;7XTCF:GFP)*, *Tg(scxa:mCherry; hsp70l:dnTCF)* and *Tg(scxa:mCherry; hsp70l:dkk1b-GFP)* live imaging, two to four embryos of the appropriate condition were mounted in 1% low melt agarose in EM in a Glass Bottom Slide dish. EM with 4.2% tricaine (Sigma-Aldrich, A5040; Pubchem CID: 261501) was added to the dish after the agarose had solidified. This was repeated for Wnt drug treatment live imaging, except that the appropriate working concentration of Wnt agonist/antagonist was added to EM with 4.2% tricaine. A small rectangular cut was made in the agarose plug in front of the embryo heads extending back to the anterior segment of the yoke, and this section was removed from the dish. This allowed for the posterior half of each embryo to be mounted in place by the gel, but for the head to grow and develop without restriction. Imaging was conducted on a Leica SP8 Confocal Microscope using the PL APO CS2 40×/1.10 W objective.

### Wnt perturbation heat shock and drug treatment statistical analysis

Embryos with ectopic muscle attachments in Wnt signaling perturbation studies were categorized qualitatively as 'none', 'mild' or 'severe' phenotype:

'none' had no discernable ectopic muscle attachments, normal craniofacial cartilage structure development, jaw muscle length, and no noticeable ectopic muscle attachments in the jaw muscles; 'mild' phenotypes had normal craniofacial cartilage structure development, jaw muscle length and attachment region, with at least one ectopic muscle attachment branching off from primary cranial muscles; 'severe' phenotypes had dramatically shortened cartilage structure, short jaw muscle length and multiple ectopic muscle attachments to ectopic tenocytes, tendons other than their WT attachment region or to unlabeled cells. Quantification involved comparing embryos with the 'none' phenotype against embryos with the 'mild' or 'severe' phenotype as one category. Absolute quantities of embryos with each category were counted for each condition and compared using the chi square test of independence in Microsoft Excel with ns=not significant, *$P<0.05$, **$P<0.01$ and ***$P<0.001$.

## Acknowledgements

We thank Ines Gehring and Natalia Libby for care of fish lines and maintenance, Dr Melanie Oakes, Dr Valentina Ciobanu and Dr Quy Nguyen at the Genomics Research and Technology Hub for their sequencing services, Raquel Bowman and Dr Dae Seok Eom for providing the *mpeg1* transgenic fish. This study was made possible in part through access to the Optical Biology Core Facility of the Developmental Biology Center, a shared resource supported by the Cancer Center Support Grant (CA-62203) and Center for Complex Biological Systems Support Grant (GM-076516) at the University of California, Irvine.

## Competing interests

The authors declare no competing or financial interests.

## Author contributions

Conceptualization: A.S., P.K.N., D.B.D., T.F.S.; Data curation: A.S., P.K.N., C.L.M.; Formal analysis: A.S., P.K.N., C.L.M., R.R.R., J.G.C., T.F.S.; Funding acquisition: T.F.S.; Investigation: A.S., T.F.S.; Methodology: A.S., P.K.N., C.L.M.; Project administration: T.F.S.; Resources: T.F.S.; Software: P.K.N.; Supervision: T.F.S.; Validation: A.S., P.K.N., D.B.D., R.R.R., J.G.C.; Visualization: A.S., P.K.N., C.L.M., D.B.D.; Writing – original draft: A.S., P.K.N., J.G.C., T.F.S.; Writing – review & editing: A.S., P.K.N., D.B.D., R.R.R., J.G.C., T.F.S.

## Funding

This work was supported by National Institutes of Health grants R01DE30565, R01AR67797 and R01DE13828 to T.F.S., R35DE027550 and R01DE033893 to J.G.C., National Science Foundation grant MCB2028424 to T.F.S., and by a National Science Foundation Simons Center for Multiscale Cell Fate fellowship to P.K.N. Open Access funding provided by University of California Irvine. Deposited in PMC for immediate release.

## Data and resource availability

All relevant data and details of resources can be found within the article and its supplementary information.

## Peer review history

The peer review history is available online at https://journals.biologists.com/dev/lookup/doi/10.1242/dev.205047.reviewer-comments.pdf

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
