## [Peer Review File · Development (Cambridge, England)]

Analysis of cranial tenocyte heterogeneity reveals a role for Wnt signaling in tendon attachments

Arul Subramanian, Pavan K. Nayak, Cameron Lee Miller, Daniel B. Dranow, Ryan Roberts, J. Gage Crump and Thomas F. Schilling
DOI: 10.1242/dev.205047

Editor: Steve Wilson

Review timeline

Original submission:	20 June 2025
Editorial decision:	11 August 2025
First revision received:	12 November 2025
Accepted:	15 December 2025

Original submission

First decision letter

MS ID#: dev.205047

MS TITLE: Analysis of cranial tenocyte heterogeneity reveals a role for Wnt signaling in tendon attachments

AUTHORS: Arul Subramanian, Pavan K. Nayak, Cameron Lee Miller, Daniel B. Dranow, Ryan Roberts, J. Gage Crump and Thomas F. Schilling

Dear Tom,

I have now received all the referees' reports on the above manuscript, and have reached a decision. The referees' comments are appended below, or you can access them online: please go to:

As you will see, the referees express considerable interest in your work, but have some suggestions for improvements. If you are able to revise the manuscript along the lines suggested, I will be happy receive a revised version of the manuscript. Please also note that Development will normally permit only one round of major revision. If it would be helpful, you are welcome to contact us to discuss your revision in greater detail. Please send us a point-by-point response indicating your plans for addressing the referees' comments, and we will look over this and provide further guidance.

Please attend to all of the reviewers' comments and ensure that you clearly highlight all changes made in the revised manuscript. Please avoid using 'Tracked changes' in Word files as these are lost in PDF conversion. I should be grateful if you would also provide a point-by-point response detailing how you have dealt with the points raised by the reviewers in the 'Response to Reviewers' box. If you do not agree with any of their criticisms or suggestions please explain clearly why this is so.

Reviewer 1

SUMMARY OF THE ADVANCE MADE IN THIS PAPER AND ITS POTENTIAL SIGNIFICANCE TO THE FIELD

Subramanian and colleagues describe the molecular heterogeneity with corresponding anatomical identification of tendon-forming cells in the zebrafish head. Using single cell RNAseq methods and

high resolution in situ hybridization they define subsets of cells corresponding to distinct cranial structures and for locations within tendons. Pathway analysis revealed differences in Wnt signaling in subsets of tenocytes, and genetic and pharmacological manipulation of the Wnt pathway alters muscle attachments to tendons. This work provides a new basis for classification of these important cell types and a resource for further studies.

SUGGESTIONS TO AUTHORS

1. Overall, cluster analysis would perhaps be easier to follow if clusters numbers, identifications and gene expression were represented by dot plots rather than umap feature plots. It is challenging to synthesize and compare expression across figures, e.g. fig 2 vs supplemental figs 1-4. Moreover Supplements 1 and 2 are ungrounded to any cluster map. Additionally a summary table of anatomical structure, abbreviation, cluster number and distinguishing markers would be helpful.

2. I understand the value of subclustering, but at present the value of the lower-resolution clustering in Figure 1 is unclear. Why not just start with the analysis of figure 2? For example, Cluster 0 is labeled Tenocytes-1 but by core expression (Fig 1E) this cluster appears to be very low expressing. Is this being driven by a subset of the cells? Double-positive cells (Fig 1H, I) seem to already distinguish subregions of each cluster 3 and 7.

3. The subcluster analysis for enthesis and MTJ designations is interesting but difficult to follow as presented. For Figures 3 and 4, it is challenging to assess how the selected markers are distributed across subclusters. UMAPS in Fig 3 E,L are too small to distinguish. In Figure 3 it looks like there is a different colormap for expression levels compared to figs 2 and 4 (but no legend is provided). In Figure 4 it is hard to determine that the expression of different markers across each umap. Figure 5 is a better representation.

4. In addition it is not clear what was done to identify new markers for enthesis and MTJ. Were clusters 35 (Fig 3), 30 (Fig 4) and 19 (fig 5) subjected to reclustering and differential expression analysis to identify new markers? But they appear to be shown using the original umap coordinates.

5. In general, more detail is needed about how the differential gene analysis was done. Give that there were biological replicates, was this taken into account for the analysis? It is not clear why certain genes were selected for further analysis.

6. Similarly it is not clear how gene modules (Table 1) were identified. Is this by prior knowledge? Co-expression?

7. Pathway analysis (Fig 7) suggests that subsets of cells (12, 16, 19, 29) are influenced by Wnt signaling while others (30, 32, 35, 36) are not. It would be nice if this was more directly tested. One would then predict that the first group of cells would express the Wnt activity reporter (Fig 8) and would be altered by modulation of signaling (Figs 9 and 10). In addition one would predict that the second group cells would not express the activity reporter and might not be influenced by signal modulation.

Other comments:

Figure 1 A. There should be a legend on the figure for the color code.

Figure 1 E. What set of genes? Is the core tendon genes module the same as the Tendon Core Module of Table 1? What is the scale?

Figure 1 F. Top 10 genes should be included as a table. Tenocytes-3 label is used twice.

Figure 3. There is no legend description for panels M-O and there are two extra unlettered panels (presumably P and Q?)

Fig 6. Which clusters have been grouped in A and B? What do the numbers (16_1 etc) represent? Are these subclusters?

Figure 7 - Anatomical identity label would be useful in addition to cluster number. Panel B - why are some cluster numbers included and others not in each panel? What does the heatmap represent?

Figure 8. It would be helpful to have the combined channels for insets displayed first to aid in orientation (e.g. switch 8B and 8C; G and H).

Figure 9, 10. Arrows or other annotations would aid in identifying the key changes to focus on. Details for statistical testing, N should be included in the legend for Fig 9 C,L and Fig 10C.

Reviewer 2

SUMMARY OF THE ADVANCE MADE IN THIS PAPER AND ITS POTENTIAL SIGNIFICANCE TO THE FIELD

In their manuscript "Analysis of cranial tenocyte heterogeneity reveals a role for Wnt signaling in tendon attachments" Subramanian and colleagues use single cell multiomics to characterize the diversity of *scxa*-positive lineages in the zebrafish head. They sorted for *scxa*:mCherry-positive cells at 72 hpf and processed for scRNA-seq. Using Seurat they identified 16 clusters, 9 of these clusters express *scxa*, *mkxa*, *tnmd*, *thbs4b* and *tnc*, well established tenocyte markers. They went on to increase cluster resolution, ending up with 42 clusters, which they show that at least a subset of these represent individual tendon populations. They then subclustered based on functionality, softer versus harder attachments and found functionally distinct subpopulations of cells related to enthesal and MTJ tenocytes that vary depending upon the hardness of the attachment. They combined tendon clusters together and did the same for ligaments. They analyzed these combined clusters and a joint cluster to compare ECM gene expression across these connective tissues. Similar analyses were performed to compare entheses to MTJ. In both sets of analyses they found differences in overall expression. They analyzed pathway enrichment across clusters and identified *Tgfb* (as would be expected) and Wnt signaling. Using Wnt signaling transgenics they demonstrate that a subpopulation of tenocytes are Wnt responsive. Variable muscle defects were observed in DnTCF transgenics, largely effecting the IMA, IMP, IH and HH muscles, with the IMP appearing most severe. Similar, albeit less severe, results were obtained in *dkk1b* expressing transgenics. Interestingly, they find similar effects via chemical inhibition or activation of the Wnt pathway. The bioinformatic analyses provide a great step forward in our understanding of tendon and ligament diversity and will be of great importance for those studying craniofacial or tendon development. The functional analyses provide evidence that the bioinformatics predicts functionally relevant pathways critical for tendon development. The manuscript is well written and the analyses are rigorous.

SUGGESTIONS TO AUTHORS

My specific concerns are:

- 1) While the *scxa*:mCherry line is well characterized and appears to faithfully recapitulate endogenous express patterns, these analyses have not been performed at the single cell level. The authors should determine the percentage of cells in their dataset that are *scxa*-positive. The authors posit that some clusters that they observe are due to contaminant cells, such analyses would provide insight into this possibility. How many of the 7 clusters that did not express the tenocyte markers may have never been *scxa*-positive? This may not be an easy question to answer given the likely perdurance of mCherry protein relative to *scxa* mRNA. However, the extent to which this is possible should be characterized and discussed in the manuscript.
- 2) Figure 1 H and I would benefit from some sort of annotation (such as arrowheads) to assist the reader in following what the authors are describing (e.g. where is Meckel's cartilage?). Also Figure 1I appears to be referenced as 1G in the text.
- 3) Figure 1 A and B are difficult to read. Perhaps if tendons and ligaments were presented in different panels it would be easier to read.

4) The authors state that unique signaling pathways across spatially distinct tendons have not been well defined. However, McGurk et al., 2017 does demonstrate this for the RA pathway and is unreferenced.

Other minor concerns:

1) "...as well as platelet derived growth factor receptor b, pdgfra..." Do the authors mean the receptor, pdgfra, or the ligand pdgfra?

Reviewer 3

SUMMARY OF THE ADVANCE MADE IN THIS PAPER AND ITS POTENTIAL SIGNIFICANCE TO THE FIELD

In this manuscript, Subramanian*, Nayak* et al. performed single cell RNA-sequencing of cranial scxa+ cells (tenocytes and ligamentocytes) during zebrafish development (at 72 hours post-fertilization). Using this dataset, the authors resolve distinct transcriptional profiles of tenocytes and ligamentocytes from tendons/ligaments located in different spatial positions within the developing pharyngeal arches as well as tenocytes at tendon attachment sites (myotendinous vs. entheseal). In addition, the authors elucidate differential transcriptional ECM profiles which correlate with tendon/ligament function and the nature of attachment sites (stiffer vs. softer). Finally, they identify a role for Wnt signaling in myotendinous junction development in a subset of cranial tendons. The work is clearly presented and will be a valuable resource that would be of great interest for the connective tissue and broader developmental biology communities.

SUGGESTIONS TO AUTHORS

Major comments:

1. The authors claim to have identified a scxa+/mpeg1+ resident tendon macrophage population. Based on the data, it is clear there is a macrophage population in close proximity to tenocytes, however it is less obvious whether these are scxa+ or not. Based on the single cell data, the macrophage cluster is only positive for mCherry. This could definitely be due to low levels of scxa transcripts, however it is also possible that macrophages engulf scxa:mCherry+ cells. If this is the case, it is possible that macrophages could become mCherry+ and that they do not truly express scxa. In the referenced work from Bautista et al. (2023), F4/80+ macrophages which were tendon resident did not express tendon reporters (ScxGFP or Scx-cre-TdTom-lineage). Therefore, the authors should provide further evidence to support this claim by either performing live imaging at higher resolution and/or performing double in situ for scxa and macrophage markers.
2. The authors mention that they identified 9 scxa+ tenocyte clusters; however, Figure 1F is confusing in terms of labeling of the clusters on the heatmap. "Tenocytes - 3" is repeated twice. In addition, clusters 11 and 12 are referred to as "Tenocytes - 6" and "Tenocytes - 7" but do not appear to express the core tendon markers and express only mCherry based on the violin plots. The authors should discuss what these clusters may be and also revise the labeling of the tenocyte clusters on the heatmap to be more intuitive.
3. Given the high expression of the Wnt reporter in non-tenocyte populations including cartilage during development, the authors could speculate more as to the potential non-cell autonomous roles Wnt may play on tendon development in their discussion section. In addition, it would be interesting to speculate on why Wnt seems to be important for this specific subset of tendons during development.
4. Were there any transcripts specifically expressed in only tendons or ligaments?
5. The images in Figure 3 and 4 would benefit from nuclear counterstaining to make it easier for the reader to orient themselves (in particular for the higher magnification insets).

Minor comments:

1. On page 10, the statement regarding the co-expression of comp and tbx1 in tenocytes in the HH-HH tendon for Figure 2I is redundant with the prior paragraph. Authors should remove either this statement or the prior one.
2. When referring to tenocyte-resident macrophage populations on page 10, Supplemental Figure 4D, not 3D should be referenced. Similarly, the supplementary figure references for the

periligamentous and peritendinous populations as well as the proliferating tenocyte module scoring is incorrect as well. Please change this accordingly.

3. Supplementary Figure 7 should be 6 in the text on page 17. Also, the captions appear to be swapped in the supplemental materials.

4. Supplementary Figure 4 reference should be 8 in the text on page 17

5. The statement on page 21 at the end of the first paragraph stating that "While we observe... MTJ shared similar expression patterns suggesting conserved ECM organization at the MTJ irrespective of the attachment stiffness at the enthesis" is contradicted by the beginning of the paragraph on page 23: "Tenocytes at the MTJ also show unique ECM expression profile that correlates with the stiffness of the tendon attachment. Please edit this accordingly.

First revision

Author response to reviewers' comments

REVIEWER 1:

1. Overall, cluster analysis would perhaps be easier to follow if clusters numbers, identifications and gene expression were represented by dot plots rather than umap feature plots. It is challenging to synthesize and compare expression across figures, e.g. fig 2 vs supplemental figs 1-4. Moreover Supplements 1 and 2 are ungrounded to any cluster map. Additionally a summary table of anatomical structure, abbreviation, cluster number and distinguishing markers would be helpful.

We have included dot plots to accompany the Nebulosa plot density plots in Figure 7 and Supplementary Figures 2, 4, 5, 6. We have also added a table summarizing the abbreviations used for tendons and ligaments, as well as relevant clusters (where identified) and corresponding markers, (Table 1).

2. I understand the value of subclustering, but at present the value of the lower-resolution clustering in Figure 1 is unclear. Why not just start with the analysis of figure 2? For example, Cluster 0 is labeled Tenocytes-1 but by core expression (Fig 1E) this cluster appears to be very low expressing. Is this being driven by a subset of the cells? Double-positive cells (Fig 1H, I) seem to already distinguish subregions of each cluster 3 and 7.

We have included the UMAP at a resolution of 0.5 in Figure 1, to show a hierarchy of clustering, i.e. that the cells initially cluster at low resolution according to differential expression of established craniofacial patterning genes such as *hand2*, *dlx* genes, etc. Then, to further identify spatially distinct tenocyte clusters, we clustered cells at a higher resolution of 1.2 (Figure 2), and identified additional differentially expressed genes that mark tendons in distinct locations, such as the medial ch-SH (ceratohyal-sternohyoideus tendon) and EOMTs (extraocular muscle tendons) (Figure 2). Lastly, the subclustering of individual spatially distinct clusters show functional domains specific to each tendon (i.e. MTJ vs enthesis), shown in Fig. 3, 4, 5, with the unique ECM patterns to these domains shown in Fig. 6. The tendon core module shows a weighted expression of multiple genes and may be skewed by the expression of individual genes. Both *scxa* and *mCherry* are strongly expressed in tenocyte clusters, but show some differences in expression levels in some clusters (See Supplementary Figure 1). This may be due to differences in expression of the *scxa* BAC-driven *mCherry* transgene versus native expression based on number of insertions or local effects at the sites of insertions.

3. The subcluster analysis for enthesis and MTJ designations is interesting but difficult to follow as presented. For Figures 3 and 4, it is challenging to assess how the selected markers are distributed across subclusters. UMAPs in Fig 3 E,L are too small to distinguish. In Figure 3 it looks like there is a different colormap for expression levels compared to figs 2 and 4 (but no legend is provided). In Figure 4 it is hard to determine that the expression of different markers across each umap. Figure 5 is a better representation.

We have adopted a uniform color scheme across all the nebulosa density plots and added violin plots for individual subclusters in Fig. 3, 4, 5 to better display marker distribution. We have also redesigned the figures to improve visualization of the plots and have included dot plots wherever necessary to help with interpretation of the data. We have also included legends for all nebulosa and dot plots.

4. In addition it is not clear what was done to identify new markers for entheses and MTJ. Were clusters 35 (Fig 3), 30 (Fig 4) and 19 (fig 5) subjected to reclustering and differential expression analysis to identify new markers? But they appear to be shown using the original umap coordinates.

We apologize for the confusion. Clusters 35, 30, and 19 were subclustered based on differential gene expression analysis but a new UMAP calculation was not performed. Our reasoning was that we needed to keep these subclusters within the context of the entire dataset so that module-score calculations of MTJ and Entheses in Figure 6C could be cross-comparable. We mentioned in the Results section that we “reclustered”, and this may have led to the confusion. We have edited the Results section of the manuscript to clarify this.

5. In general, more detail is needed about how the differential gene analysis was done. Give that there were biological replicates, was this taken into account for the analysis? It is not clear why certain genes were selected for further analysis.

We integrated the data from three biological replicates followed by cell clustering at selected resolution. This allowed for batch-correction across samples for UMAP visualization. However, differential expression analysis was not performed on integrated data, as this is not statistically valid. We used the standard FindAllMarkers() function that performs a Wilcoxon rank sum test to analyze differential gene expression using log-normalized RNA counts across all replicates. We explored these differentially expressed genes manually to look for markers that showed the most specific expression for individual UMAP clusters and selected these genes for experimental validation via *isHCRs*. To verify that batch effects were not affecting clustering, we split the dataset by biological replicate and observed the distributions of these selected marker genes across all clusters (see Figure 2 and figures below). Individual marker genes had high expression (high z-scores) that matched their validated cluster in Figure 2, and this was repeated across each replicate with only slight deviations.

6. Similarly it is not clear how gene modules (Table 1) were identified. Is this by prior knowledge? Co-expression?

We created the gene modules using a combination of HCR-verified genes from this study, peer-reviewed studies in zebrafish and other model systems. We have updated the references and module table (Supplementary Table 1) to show these studies.

7. Pathway analysis (Fig 7) suggests that subsets of cells (12, 16, 19, 29) are influenced by Wnt signaling while others (30, 32, 35, 36) are not. It would be nice if this was more directly tested. One would then predict that the first group of cells would express the Wnt activity reporter (Fig 8) and would be altered by modulation of signaling (Figs 9 and 10). In addition one would predict that

the second group cells would not express the activity reporter and might not be influenced by signal modulation.

We thank the reviewer for this suggestion. Our KEGG pathway analysis showed Wnt pathway gene expression in clusters 12 (mc-mc ligament), and 29 (HH-HH). Further evidence included colocalization of GFP and mCherry signals in the double transgenic line 7Xtcf:GFP; scxa:mCherry, in mc-mc ligament (Fig 8K-O) and HH-HH (Fig 8U-X) tendon. In the same embryos, we did not observe colocalization of GFP and mCherry signals in joint (cluster 32) or ch-SH tenocytes (cluster 35). Heatshock experiments also support this hypothesis as the defects and ectopic attachments were restricted to HH-HH tenocytes (cluster 29) and condensation at the mc-mc ligament. We also limited the period of heatshock to stages just before these cranial tendons form. Lack of transgenics to perturb Wnt expression specifically in tenocytes or surrounding cells has hampered our efforts to address specific roles in these cells but we now acknowledge this in the Discussion - "One caveat of the study is that since all the Wnt disruptions (heatshock and drug treatments) were global, they fail to provide insight on the tissue-specific role of Wnt signaling in tendon patterning and the function of Wnt signaling in tendon development." This would indeed be an interesting avenue of future research that we can now pursue using for example scxa enhancers.

Minor comments:

Figure 1 A. There should be a legend on the figure for the color code.

We have added a legend to the figure clarifying the color code.

Figure 1 E. What set of genes? Is the core tendon genes module the same as the Tendon Core Module of Table 1? What is the scale?

Yes, the tendon gene module is the same as tendon core module of Table 1. We have now edited the titles to include the same wording. We have also included the missing scale.

Figure 1 F. Top 10 genes should be included as a table. Tenocytes-3 label is used twice.

We have now included supplementary tables with the top10 genes for the clustering in both figure 1 and 2 (see supplementary tables 1 and 3). We have corrected the Tenocytes-3 label._

Figure 3. There is no legend description for panels M-O and there are two extra unlettered panels (presumably P and Q?)

We have included the legends to the panels with plot densities. The unlettered panels belong to the single panel. A box has been added to group the panels and help interpretation.

Fig 6. Which clusters have been grouped in A and B? What do the numbers (16_1 etc) represent? Are these subclusters?

To answer the first question, we mentioned in the results section "Module scores were compared across tissue types by grouping clusters together. Clusters 19 (EOMTs), 29 (HH-HH), 16 (Pec fin tendons), 35 (ch-SH), 30 (Dermal bone tendons) were combined to form a tendon cluster, clusters 12 (mc-mc) and 36 (mc-pq-hs) combined to form a ligament cluster, cluster 32 (mc-pq, mc-mc, ch-hs) comprised a joint cluster (with associated tenocytes) (Fig. 6A)." For Figure 6B, we have now removed the numbers which previously referred to subclusters of individual clusters from Fig. 2. We have now stated "We also observed distinct ECM expression profiles between tenocytes of Enthesis (grouped Pec fin, Dermal bone, ch-SH, and EOMT entheses subclusters) versus MTJ (grouped Pec fin, Dermal bone, ch-SH, EOMT and HH-HH MTJ subclusters) (Fig. 6B). We have added these groupings to the Figure 6 legend as well to avoid any further confusion. .

Figure 7 - Anatomical identity label would be useful in addition to cluster number. Panel B - why are some cluster numbers included and others not in each panel? What does the heatmap represent?

We have now included the anatomical names of tendons and ligaments associated with the cluster numbers and have provided a legend for the dot plot.

Figure 8. It would be helpful to have the combined channels for insets displayed first to aid in orientation (e.g. switch 8B and 8C; G and H).

We have revised the orientation and insets in Fig 8.

Figure 9, 10. Arrows or other annotations would aid in identifying the key changes to focus on. Details for statistical testing, N should be included in the legend for Fig 9 C,L and Fig 10C.

We have added arrows to guide the readers to specific tendons or tenocytes. We also include the N values in the legends for Fig 9 and 10 and supplementary figures.

REVIEWER 2:

1) While the *scxa*:mCherry line is well characterized and appears to faithfully recapitulate endogenous express patterns, these analyses have not been performed at the single cell level. The authors should determine the percentage of cells in their dataset that are *scxa*-positive. The authors posit that some clusters that they observe are due to contaminant cells, such analyses would provide insight into this possibility. How many of the 7 clusters that did not express the tenocyte markers may have never been *scxa*-positive? This may not be an easy question to answer given the likely perdurance of mCherry protein relative to *scxa* mRNA. However, the extent to which this is possible should be characterized and discussed in the manuscript.

In Supplementary Figure 1A, we have calculated the cell numbers and distribution as requested and provided the results in a new Supplementary Figure 1. 67.44% of the cells were *mCherry* positive, while 44.21% of cells were *scxa* positive. 37% of cells expressed both *mCherry* and *scxa*. Further, we examined the distributions of percentage of cells which have >0 *scxa* and >0 *mCherry* reads and found that all clusters contained *scxa*⁺ and *mCherry*⁺ cells (Supplementary Fig. 1B-C).

With respect to the clusters that expressed very low levels of *scxa*, we cannot speculate if they would have expressed *scxa* at an earlier timepoint. We have also considered the perdurance of mCherry protein relative to *scxa* RNA that could have led to capture of these low *scxa* expressing cells. This could also be due to low read capture efficiency inherent to 10X sample preparation.

2) Figure 1 H and I would benefit from some sort of annotation (such as arrowheads) to assist the reader in following what the authors are describing (e.g. where is Meckel's cartilage?). Also Figure 1I appears to be referenced as 1G in the text.

We have now added arrowheads to help guide the reader to the tendons and ligaments being discussed in the results. Cartilages were not indicated since these embryos were not stained with a cartilage marker. We also corrected the reference to the correct panel in Fig 1.

3) Figure 1 A and B are difficult to read. Perhaps if tendons and ligaments were presented in different panels it would be easier to read.

Figure 1 has been simplified somewhat and we have edited the image to show ligaments and tendons in different colors to make them easier to distinguish.

4) The authors state that unique signaling pathways across spatially distinct tendons have not been well defined. However, McGurk et al., 2017 does demonstrate this for the RA pathway and is unreferenced.

We have now revised the statement and added the missing reference.

Minor comments

1) "...as well as platelet derived growth factor receptor b, pdgfra..." Do the authors mean the receptor, pdgfra, or the ligand pdgfra?

We meant the ligand not the receptor, as this is what is referenced as a macrophage marker in the Bautista et al. 2023 study we have referenced. We have revised this error in the manuscript.

REVIEWER 3

1. The authors claim to have identified a *scxa*⁺/*mpeg1*⁺ resident tendon macrophage population. Based on the data, it is clear there is a macrophage population in close proximity to tenocytes, however it is less obvious whether these are *scxa*⁺ or not. Based on the single cell data, the macrophage cluster is only positive for mCherry. This could definitely be due to low levels of *scxa* transcripts, however it is also possible that macrophages engulf *scxa*:mCherry⁺ cells. If this is the case, it is possible that macrophages could become mCherry⁺ and that they do not truly express *scxa*. In the referenced work from Bautista et al. (2023), F4/80⁺ macrophages which were tendon resident did not express tendon reporters (*Scx*GFP or *Scx*-cre-TdTom-lineage). Therefore, the authors should provide further evidence to support this claim by either performing live imaging at higher resolution and/or performing double in situ for *scxa* and macrophage markers.

We attempted to verify if *scxa* is expressed in *mpeg1*-eYFP⁺ macrophages with antibody staining but were unable to successfully stain the transgenic macrophages and verify coexpression of *scxa*. Hence we have edited the results and have removed any mention of this macrophage population as being a "tendon-resident macrophage" population.

2. The authors mention that they identified 9 *scxa*⁺ tenocyte clusters; however, Figure 1F is confusing in terms of labeling of the clusters on the heatmap. "Tenocytes - 3" is repeated twice. In addition, clusters 11 and 12 are referred to as "Tenocytes - 6" and "Tenocytes - 7" but do not appear to express the core tendon markers and express only mCherry based on the violin plots. The authors should discuss what these clusters may be and also revise the labeling of the tenocyte clusters on the heatmap to be more intuitive.

We have corrected the labeling error. The core tendon module creates a weighted expression feature plot, but does not show individual tenocyte genes. We see changes in expression of different tendon core markers in some tenocytes. Clusters 11 (Tenocytes-7) and 12 (Tenocytes-8) were labelled as tenocyte clusters because they contained significant numbers of cells expressing the core markers. We have yet to identify which tendons these represent anatomically and we have only focused on ventral jaw tendons. We now mention this limitation of the study in Discussion.

3. Given the high expression of the Wnt reporter in non-tenocyte populations including cartilage during development, the authors could speculate more as to the potential non-cell autonomous roles Wnt may play on tendon development in their discussion section. In addition, it would be interesting to speculate on why Wnt seems to be important for this specific subset of tendons during development.

We agree that it would be nice to address roles for Wnt signaling directly in cranial tenocytes as discussed in response to Reviewer 1's comment #7. The *7Xtcf*:GFP Wnt reporter is only expressed in a subset of embryonic tenocytes and similar subsets of tendons are affected by multiple different Wnt perturbations. Our lab has also previously shown that Wnt signaling regulates migration and condensation of neural crest cells (Alexander et al. and Wnt signaling non-autonomously affects mesodermal differentiation and mesenchyme proliferation in developing mouse limbs (ten Berge et al. 2008).). Since wnt is a secreted ligand, we hypothesize that only tenocytes that express the required cognate receptors can respond to wnt signaling. The formation of mc-mc ligament involves migration and condensation at the Meckel's cartilage and wnt signaling could regulate the process effectively. Wnt may also play a role in balancing other cues to stabilize the fate of specific tendons, as has been seen in limb tendons where modulating Wnt and TGF β signaling can lead to dynamic effects on *Scx* expression (Garcia-Lee et al. 2021). We have mentioned caveats to interpreting global perturbations of Wnt signaling in our discussion.

4. Were there any transcripts specifically expressed in only tendons or ligaments?

We identified some tendon-specific markers including *comp* (*thbs5*) (See Supplementary Figure 3N-Q), and *mkxb* which only marked the ch-SH (See Figure 2C and Figure 3). We did not identify any ligament-specific markers, but did note that *mkxa*, *tnmd*, and *fndc1* had strong ligament expression, while our companion paper by Roberts et al. identified *ognb* as a more specific, marker of ligaments (particularly the mc-pq and pq-hs ligaments) (see Fig 4B & D in Roberts et al.).

5. The images in Figure 3 and 4 would benefit from nuclear counterstaining to make it easier for the reader to orient themselves (in particular for the higher magnification insets).

We had previously tried to add the nuclear signal in the image and it clouded the signal from the probes. So we have instead opted to include the 3D masks of the nuclei showing their volume.

Minor comments

1. On page 10, the statement regarding the co-expression of *comp* and *tbx1* in tenocytes in the HH-HH tendon for Figure 2I is redundant with the prior paragraph. Authors should remove either this statement or the prior one.

This statement has been removed.

2. When referring to tenocyte-resident macrophage populations on page 10, Supplemental Figure 4D, not 3D should be referenced. Similarly, the supplementary figure references for the periligamentous and peritendinous populations as well as the proliferating tenocyte module scoring is incorrect as well. Please change this accordingly.

The figure references have been fixed.

3. Supplementary Figure 7 should be 6 in the text on page 17. Also, the captions appear to be swapped in the supplemental materials.

These references and captions have also been fixed.

4. Supplementary Figure 4 reference should be 8 in the text on page 17

This reference has been fixed.

5. The statement on page 21 at the end of the first paragraph stating that "While we observe... MTJ shared similar expression patterns suggesting conserved ECM organization at the MTJ irrespective of the attachment stiffness at the enthesis" is contradicted by the beginning of the paragraph on page 23: "Tenocytes at the MTJ also show unique ECM expression profile that correlates with the stiffness of the tendon attachment. Please edit this accordingly.

We have now removed the original statement on page 21, and have edited the second sentence on page 23 to mention "Tenocytes at the MTJ also show unique ECM expression profiles across spatially distinct tendons."

Second decision letter

MS ID#: dev.205047R1

MS TITLE: Analysis of cranial tenocyte heterogeneity reveals a role for Wnt signaling in tendon attachments

AUTHORS: Arul Subramanian, Pavan K. Nayak, Cameron Lee Miller, Daniel B. Dranow, Ryan Roberts, J. Gage Crump and Thomas F. Schilling

ARTICLE TYPE: Research Article

Dear Tom,

The referees are happy with your revisions and so I am happy to tell you that your manuscript has been accepted for publication in Development, pending our standard publication integrity checks.

Reviewer 1

The authors have addressed my previous concerns. Data are more clearly presented and additional information provided will be helpful to the reader.

Reviewer 2

SUMMARY OF THE ADVANCE MADE IN THIS PAPER AND ITS POTENTIAL SIGNIFICANCE TO THE FIELD

SUGGESTIONS TO AUTHORS

The authors have addressed my previous concerns.

Reviewer 3

SUMMARY OF THE ADVANCE MADE IN THIS PAPER AND ITS POTENTIAL SIGNIFICANCE TO THE FIELD

The authors have addressed all of my previous comments on their manuscript. It is a thorough work establishing the diversity of tendon cell populations during craniofacial development and presents a novel role for Wnt signaling in regulating tendon patterning. I believe their work will be of high interest and a valuable resource to the community.